# Decoupled Low-Rank Adaptation for Robust Federated Fine-Tuning

Xiuwen Fang [* 1]  Xuliang Yang [* 1]  Mang Ye [1]

## Abstract

Federated Learning (FL) enables collaborative training across distributed clients while preserving data privacy. However, fine-tuning large-scale pre-trained models in FL is hindered by resource constraints and communication costs. Although introducing parameter-efficient fine-tuning strategies such as Low-Rank Adaptation (LoRA) effectively reduces trainable parameters, this low-rank constraint exacerbates noise sensitivity, leading to overfitting and aggregation bias. Existing robust federated fine-tuning methods rely on additional proxy data and treat low-rank adapters as generic weight vectors. In this paper, we investigate the structural properties of LoRA and reveal a robustness asymmetry. The down-projection matrix $A$ extracts stable general features, whereas the up-projection matrix $B$ is highly susceptible to fitting noise patterns. Based on this finding, we propose Federated Decoupled Robust LoRA (FedDR-LoRA), which employs a dual-branch mechanism to decouple robust feature learning from noise modeling and mitigates noise interference through noisy branch negative learning. During federated aggregation, we establish global consensus through aggregating $B$ while preserving local feature alignment in $A$. Extensive experiments demonstrate that FedDR-LoRA outperforms existing state-of-the-art methods across various noisy federated scenarios. Our code is available at: https://github.com/FangXiuwen/FedDR-LoRA.

## 1. Introduction

Federated Learning (FL) (McMahan et al., 2017) is a paradigm that enables collaborative model training on distributed edge devices while preserving data privacy. In a standard FL process, participating clients download the global model, perform local training on their private datasets, and then upload the model updates to a central server for aggregation, thereby iteratively optimizing the global model. Recently, large-scale Pre-Trained Models (PTMs) demonstrate remarkable generalization capabilities, laying the foundation for various downstream tasks (Kuang et al., 2024). PTMs provide a powerful knowledge base, thereby improving the performance of data-scarce clients in FL. FL enables privacy-preserving adaptation of PTMs from distributed datasets. Therefore, deploying PTMs in FL becomes a key strategy for deploying capable models on edge devices (Zhang et al., 2023b). However, large-scale PTMs are constrained by the limited computational and storage resources of edge devices. Besides, the iterative transmission of full parameters in FL incurs substantial communication overhead. To mitigate these bottlenecks, Parameter-Efficient Fine-Tuning (PEFT) techniques are widely used for edge deployments, which enable PTMs fine-tuning by updating a small subset of trainable parameters.

Among various PEFT approaches, Low-Rank Adaptation (LoRA) (Hu et al., 2022) is widely adopted for federated deployment, as it maintains performance comparable to Full Fine-Tuning (FFT) while updating only a small number of parameters. However, deploying LoRA in practical federated systems is often hampered by the pervasive problem of label noise. In a federated environment, high-quality clean data is often difficult to obtain (Ye et al., 2023; Jiang & Zhang, 2025). Limited local expertise inevitably leads to mislabeled data. To protect local privacy, some clients may intentionally introduce noisy data. While LoRA effectively compresses the trainable parameter space, this low-rank constraint exacerbates its sensitivity to label noise (Fang & Ye, 2025). The compressed parameter manifold forces the model to overfit noisy data, leading to significant weight divergence during local training and causing harmful aggregation bias, ultimately impairing the generalization ability of the global model (Li et al., 2020b).

Existing robust FL methods primarily target full-parameter models and do not consider the unique structural characteristics of low-rank adapters. Although recent studies such as RFedLR (Fang & Ye, 2025) attempt to mitigate this issue, they typically rely on the strong assumption of a clean proxy dataset being available on the server side for

---

[*]Equal contribution  [1]School of Computer Science, Wuhan University, Wuhan, China. Correspondence to: Mang Ye <ye-mang@whu.edu.cn>.

*Proceedings of the $43^{rd}$ International Conference on Machine Learning*, Seoul, South Korea. PMLR 306, 2026. Copyright 2026 by the author(s).

parameter sensitivity analysis, constraining their practical application. Furthermore, these methods operate in a model-agnostic manner, treating the LoRA adapter as a single weight vector, ignoring the intrinsic structural properties of the low-rank architecture. Consequently, a robust federated LoRA framework specifically designed for low-rank adapter architectures is essential.

To address this challenge, we investigate the decomposition matrix of LoRA, including the down-projection matrix $A$ and the up-projection matrix $B$, along with their distinct characteristics under noisy conditions. The down-projection matrix $A$ maps high-dimensional inputs to a low-dimensional space to extract general features of the input distribution. Since label noise slightly affects the input distribution, the features remain consistent across clean and noisy data, making $A$ robust to label noise. In contrast, the up-projection matrix $B$ maps latent features to task outputs and is directly influenced by the conditional label distribution. Thus, it is more sensitive and prone to fitting noise patterns to reduce empirical risk. This functional distinction indicates that we can leverage the robustness of the projection subspace and utilize $B$ to separate effective semantic information from noise.

Based on this insight, we propose Federated Decoupled Robust Low-Rank Adaptation (FedDR-LoRA) to achieve robust federated fine-tuning with noisy labels. First, we introduce a decoupled LoRA fine-tuning mechanism that leverages the functional asymmetry of decomposed matrices $A$ and $B$. A clean branch is designed to be optimized to integrate stable feature representations, while a noisy branch is actively guided to learn noise patterns. Second, to fully utilize noisy samples, we propose a noisy branch negative learning. We employ clean branches to provide reliable soft targets for noisy samples while forcing the prediction distributions of clean branches to diverge from those of noisy branches. Finally, we implement an asymmetric global aggregation. Considering that the down-projection matrix $A$ converges to a shared manifold, we keep it locally to maintain fine-grained feature alignment, while separately aggregating the clean and noisy up-projection matrices to enable knowledge sharing without mixing clean and noise-specific semantics. Extensive experiments demonstrate that FedDR-LoRA significantly outperforms State-Of-The-Art (SOTA) methods in noisy federated scenarios. The main contributions of this work are as follows:

- We investigate the robustness asymmetry of LoRA decomposition matrices under label noise.

- We propose a robust federated fine-tuning framework, Federated Decoupled Robust Low-Rank Adaptation (FedDR-LoRA), which includes decoupled LoRA fine-tuning and noisy branch negative learning.

- Experimental results show that FedDR-LoRA outperforms SOTA methods across various noisy FL settings.

## 2. Related Work

**Parameter-Efficient Fine-Tuning.** The primary objective of PEFT is to mitigate the computational and memory costs of adapting large-scale PTMs in resource-constrained environments. Early PEFT methods introduce a small set of trainable parameters into frozen pre-trained architectures, including Prefix-Tuning (Li & Liang, 2021) and Prompt Tuning (Lester et al., 2021), which add trainable vectors or prompts at the input, and Adapters (Houlsby et al., 2019), which insert lightweight modules into transformer layers. In contrast to these methods, LoRA (Hu et al., 2022) becomes the most widely applied fine-tuning approach due to its simplicity and empirical effectiveness. It approximates model weight updates by training two low-rank matrices $A$ and $B$. Several improved methods based on LoRA have been proposed. QLoRA (Dettmers et al., 2023) reduces memory consumption by quantizing the pre-trained model to 4-bit precision while training low-rank adapters. AdaLoRA (Zhang et al., 2023a) improves parameter efficiency by adaptively allocating ranks based on parameter importance. HydraLoRA (Tian et al., 2024) further explores an asymmetric LoRA architecture to improve fine-tuning efficiency by assigning different roles to LoRA components.

**Parameter-Efficient Fine-Tuning in FL.** The substantial communication costs and constrained local resources in FL render FFT impractical (Lee et al., 2025; Koo et al., 2025). To address this challenge, a study (Zhang et al., 2023b) evaluates several PEFT methods within the FL context. The results show that the LoRA-based FL method FedLR achieves significant advantages in both computational efficiency and model accuracy. A critical challenge inherent to FedLR is the aggregation bias caused by the nonlinearity of low-rank decomposition. To mitigate this, FFA-LoRA (Sun et al., 2024) freezes matrix $A$ and updates only matrix $B$, thus reducing aggregation distortion. RoLoRA (Chen et al., 2025) dynamically alternates which matrix is frozen across rounds. LoRA-FAIR (Bian et al., 2025) introduces a residual correction term to compensate for aggregation bias. SLoRA (Babakniya et al., 2023) utilizes singular value decomposition to decompose global updates into low-rank factors. FLoRA (Wang et al., 2024) proposes a stacked heterogeneous rank LoRA aggregation mechanism to mitigate aggregation bias. FedSA-LoRA (Guo et al., 2025) shares only matrix $A$ with the server, while keeping matrix $B$ local. FedLEASE (Wang et al., 2025) adaptively allocates and selects LoRA experts for federated fine-tuning to address heterogeneous adaptation. Although prior studies observe different roles of $A$ and $B$ in centralized or federated fine-tuning, they discuss this asymmetry from the perspectives

of shared and personalized knowledge, task interference, or heterogeneous adaptation. In contrast, our work characterizes the robustness asymmetry of $A$ and $B$ under label noise. To address the robust federated PEFT problem, RFedLR (Fang & Ye, 2025) enhances local training through sensitivity-aware backpropagation and mitigates noise propagation through adaptive LoRA aggregation. However, it relies on proxy data and treats the LoRA adapter as general model weights, without explicitly exploiting the structural difference between the two LoRA factors.

**Label Noise Learning.** The proliferation of methods for learning with label noise has addressed the critical challenge of model degradation under imperfect annotations. Initial approaches focus on designing noise-robust loss functions, such as the generalized cross-entropy (Zhang & Sabuncu, 2018), which bridges the robustness of mean absolute error and the convergence benefits of cross-entropy, and the Symmetric Loss (Wang et al., 2019), which introduces a corrective term to mitigate noise impact. Beyond loss design, sample selection strategies like MentorNet (Jiang et al., 2018) dynamically weight samples during training, while Co-teaching (Han et al., 2018) leverages dual networks to mutually filter noisy data. More advanced methods explicitly model label corruption, as seen in forward correction (Patrini et al., 2017), which estimates a noise transition matrix to correct losses. DivideMix (Li et al., 2020a) combines sample division with semi-supervised learning.

**Label Noise Learning in FL.** Robust FL primarily evolves along two fundamental research directions: client-side methods for robust local training and server-side methods for noise-resistant aggregation. Concerning the client-side direction, LSR (Jiang et al., 2022) implicitly regularizes the model during local training to prevent it from memorizing noisy labels. RHFL (Fang & Ye, 2022) mitigates the impact of noisy clients on the federated system by reweighting client contributions during the aggregation phase. FedNoRo (Wu et al., 2023) discriminates between noisy and clean clients and mitigates the impact of noisy labels through knowledge distillation. For the server-side methodology, the geometric median-based aggregation method (Li et al., 2021) improves robustness through its inherent capability to differentiate and isolate benign model updates from noisy or malicious ones. FLPhish (Li et al., 2023) designs a reputation mechanism based on Bayesian inference, which adjusts the aggregation weights of clients using their historical performance. FedClean (Jiang & Zhang, 2025) uses Gaussian mixture model to separate noisy samples and relabels them with pseudo-labels and global model predictions. Although the aforementioned methods achieve promising results, they either overlook the severe computational constraints in federated scenarios or fail to fully leverage the information provided by noisy data. Building upon these foundations, this paper proposes a more effective strategy.

## 3. Method

### 3.1. Preliminary

**Problem Setup and Notations.** We consider a FL system with $K$ clients, indexed by $k \in \{1, \ldots, K\}$. Each client $k$ holds a local private dataset $\mathcal{D}_k = \{(x_i, \tilde{y}_i)\}_{i=1}^{N_k}$, containing $N_k$ samples. $x_i$ denotes the input data, and $\tilde{y}_i \in \{1, \ldots, C\}$ denotes the potentially noisy label. The total number of samples across all clients is $N = \sum_{k=1}^{K} N_k$.

Standard FFT in FL requires updating and transmitting the entire parameter set of the backbone model, which incurs substantial communication overhead and computational costs. Therefore, we adopt LoRA to approximate the weight updates. Formally, for a pre-trained weight matrix $W_0 \in \mathbb{R}^{d_{out} \times d_{in}}$, we constrain its update $\Delta W$ by representing it as the product of two low-rank matrices $B \in \mathbb{R}^{d_{out} \times r}$ and $A \in \mathbb{R}^{r \times d_{in}}$, where the rank $r \ll \min(d_{in}, d_{out})$. During the local training process, $W_0$ remains frozen, and only the adapter matrices $A$ and $B$ are optimized. The forward pass for an input $x \in \mathbb{R}^{d_{in}}$ is formulated as:

$$h = W_0 x + \frac{\alpha}{r} BAx, \tag{1}$$

where $\alpha$ is a constant scaling factor to balance the magnitude of the update. Typically, $A$ is initialized with a random Gaussian distribution, and $B$ is initialized to zero, ensuring that $\Delta W = 0$ at the beginning of training.

In our FL setup, the pre-trained model weights $W_0$ are pre-deployed to all clients and remain frozen during fine-tuning. We define the local learnable parameters for the client $k$ as $\theta_k = \{A_k, B_k\}$, where $A_k, B_k$ are the local low-rank adapter matrices. Our objective is to collaboratively optimize the set of local trainable parameters $\Theta = \{\theta_1, \ldots, \theta_K\}$ to minimize the collective empirical risk. Formally, the objective is defined as:

$$\min_{\Theta} \mathcal{F}(\Theta) = \sum_{k=1}^{K} w_k \mathcal{L}_k(\theta_k; \mathcal{D}_k), \tag{2}$$

where $w_k = \frac{N_k}{N}$ is the weight of the client $k$, and $\mathcal{L}_k$ represents the empirical risk of the local model parameterized by $\theta_k$.

**Observation and Motivation.** Different from prior analyses that mainly relate the $A/B$ distinction to aggregation bias or heterogeneous adaptation, we investigate the distinct roles of low-rank matrices $A$ and $B$ under noisy supervision. We conduct a preliminary experiment, training LoRA adapters separately on clean and noisy datasets, and then measuring the layer-wise cosine similarity of the obtained matrices. As illustrated in Figure 1, the behavior of the down-projection matrix $A$ and the up-projection matrix $B$ in LoRA exhibits significant differences. The down-projection

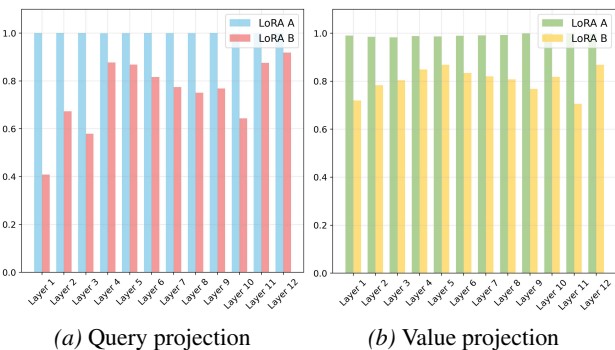

*(a) Query projection*  *(b) Value projection*

*Figure 1.* Layer-wise cosine similarity between the LoRA matrices trained with clean data and noisy data, shown for the query (left) and value (right) projections.

matrices $(A_q, A_v)$ exhibit extremely high similarity across all layers under both clean and noisy conditions, indicating that the learned input projection subspace remains virtually invariant regardless of whether the labels are correct. In contrast, the similarity scores of the up-projection matrices $(B_q, B_v)$ show significantly lower and fluctuating similarity scores, revealing that $B$ is sensitive to label noise and varies drastically to fit corrupted targets.

We attribute this phenomenon to the decomposed form $\Delta W = BA$ of the LoRA updates. The matrix $A \in \mathbb{R}^{r \times d_{in}}$ can be regarded as a feature projector, mapping high-dimensional inputs to a low-dimensional subspace to extract general feature representations from the input distribution. Label noise perturbs the conditional distribution $P(y|x)$ but does not alter the input marginal distribution $P(x)$ within each client, so $A$ can persistently learn the fundamental feature manifold. However, the matrix $B \in \mathbb{R}^{d_{out} \times r}$ functions as a semantic aligner, projecting features onto the label space. Consequently, label noise explicitly forces $B$ to distort its mapping direction to fit incorrect supervision. A mechanism-level analysis of this robustness asymmetry is provided in Appendix B, which explains how noisy supervision affects the gradients of $A$ and $B$ differently.

Building on this observation, we propose a dual-branch LoRA architecture to resolve the parameter conflict between fitting clean data and overfitting noise (Figure 2). Since a single $B$ matrix cannot simultaneously maintain a clean mapping and learn noise patterns, we decouple the up-projection process while maintaining a unified feature subspace. Specifically, each client $k$ is assigned a trainable matrix $A_k$ to capture common features, but two independent up-projection matrices are instantiated, a clean branch $B_k^c$ for fitting clean data, and a noisy branch $B_k^n$ for capturing local noise patterns. Accordingly, we denote the learnable parameters of the clean and noisy branches on client $k$ as $\theta_k^c = \{A_k, B_k^c\}$ and $\theta_k^n = \{A_k, B_k^n\}$, respectively. These two branches share the same $A_k$ matrix but maintain their own respective $B$ matrices, as shown in Figure 2. The

weight updates for the clean and noisy branches in client $k$ are $\Delta W_k^c = B_k^c A_k$ and $\Delta W_k^n = B_k^n A_k$, respectively. This design ensures that the idiosyncratic variations caused by noise are isolated within $B^n$, preserving the integrity of the robust features in $B^c$. To prevent the common feature subspace from being corrupted by label noise, $A_k$ is only updated by the clean branch and remains frozen when optimizing the noise branch.

### 3.2. Dual-Branch LoRA Warm-up

Numerous studies (Arpit et al., 2017; Han et al., 2018) reveal the memorization effect in label noise learning. Neural networks initially prioritize fitting clean data during the early stages of training. As training progresses, the interference caused by noisy data becomes increasingly pronounced. Building on this phenomenon, we posit that samples with lower loss values are more likely to be clean, whereas those with higher losses indicate potential noise.

In the warm-up phase, we employ an extreme value selection strategy to allocate training samples to each branch. Specifically, we evaluate the Cross-Entropy (CE) loss $\ell_i^c = \mathcal{L}_{ce}(f(x_i; \theta_k^c), \tilde{y}_i)$ for each sample with the clean branch in client $k$. Local clean and noisy subsets are constructed based on two quantile thresholds $\delta_{low}$ and $\delta_{high}$ of the loss distribution:

$$
\begin{aligned}
\mathcal{D}_k^c &= \{(x_i, \tilde{y}_i) \in \mathcal{D}_k \mid \ell_i^c \leq \delta_{low}\}, \\
\mathcal{D}_k^n &= \{(x_i, \tilde{y}_i) \in \mathcal{D}_k \mid \ell_i^c \geq \delta_{high}\},
\end{aligned}
\tag{3}
$$

where $\mathcal{D}_k^c$ contains low-loss, high-confidence samples for training the clean branch. $\mathcal{D}_k^n$ contains high-loss conflicting samples for training the noisy branch. Samples within the $[\delta_{low}, \delta_{high}]$ interval are considered uncertain and discarded at this stage. The specific threshold setting strategy is detailed in the experimental section.

Subsequently, we execute independent gradient updates for each branch using separate forward passes. The clean branch $\{A_k, B_k^c\}$ is optimized solely on $\mathcal{D}_k^c$ to consolidate robust decision boundaries, whereas the noisy branch $B_k^n$ is actively trained on $\mathcal{D}_k^n$ to learn noise patterns. The respective objectives are formulated as:

$$
\begin{aligned}
&\min_{\{A_k, B_k^c\}} \sum_{(x, \tilde{y}) \in \mathcal{D}_k^c} \mathcal{L}_{ce}(f(x; \theta_k^c), \tilde{y}), \\
&\min_{B_k^n} \sum_{(x, \tilde{y}) \in \mathcal{D}_k^n} \mathcal{L}_{ce}(f(x; \theta_k^n), \tilde{y}).
\end{aligned}
\tag{4}
$$

We update $A_k$ only together with $B_k^c$ on $\mathcal{D}_k^c$, and keep $A_k$ frozen when optimizing $B_k^n$ on $\mathcal{D}_k^n$. As training progresses, this divergent gradient flow ensures that $B_k^c$ evolves to represent the underlying clean data distribution, while $B_k^n$ effectively models the noise distribution.

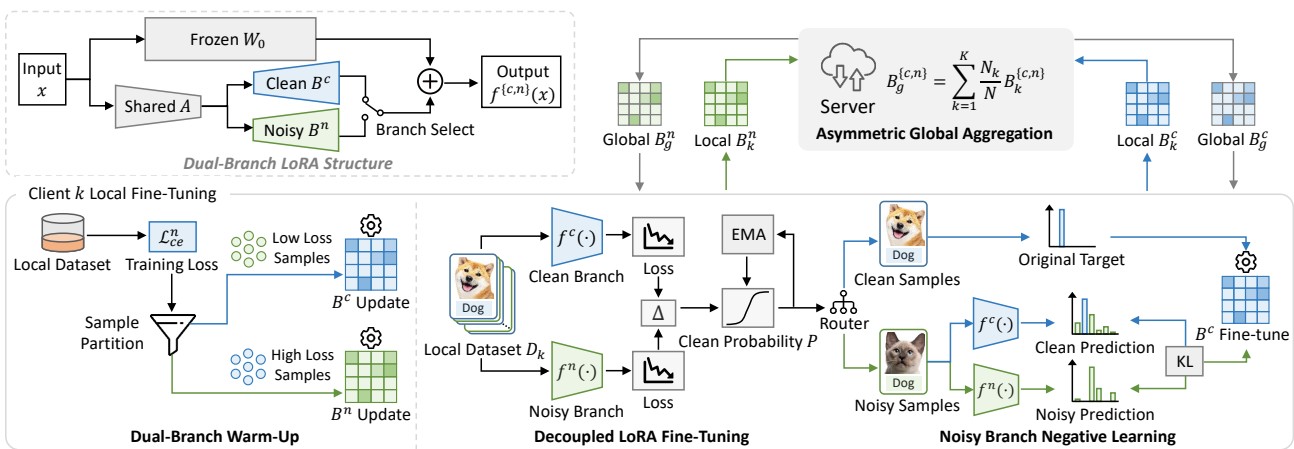

*Figure 2.* Illustration of FedDR-LoRA, which is designed for robust federated fine-tuning under label noise. During the local training phase, we introduce a decoupled LoRA fine-tuning mechanism to separate noisy samples. Then, noisy branch negative learning encourages the clean-branch predictions to diverge from the noise-branch predictions on noisy samples. In the global aggregation phase, we separately aggregate $B^c$ and $B^n$ while keeping $A$ local. The training procedure is summarized in Algorithm 1.

### 3.3. Decoupled LoRA Fine-Tuning

After initialization, we aim to enable the two branches to progressively identify their corresponding data, thereby achieving separation between clean and noisy samples. This process is driven by sample loss values. To ensure stability of the results, an Exponential Moving Average (EMA) mechanism is also incorporated.

**Discrepancy-Based Probability Estimation.** We design a collaborative training strategy driven by the prediction difference between two branches. The data allocation for different branches is dynamically determined during the training process. During the warm-up phase, the clean branch $\theta^c$ is optimized solely on low-loss samples, allowing it to capture consistent robust features. Therefore, $\theta^c$ yields low loss on clean samples. For noisy samples, the loss increases significantly due to the mismatch between their labels and the learned semantics. Conversely, the noisy branch $\theta^n$, due to its exposure to high-loss outliers, effectively memorizes the noisy patterns. Thus, for a given sample, if the loss from the clean branch is significantly lower than that of the noisy branch, it indicates that the sample conforms to the robust distribution. Conversely, if its loss is lower on the noisy branch, the sample likely belongs to the noisy distribution (Yuan et al., 2025). Therefore, the loss difference between the two branches can be utilized to measure the clean probability.

Specifically, for the client $k$ at epoch $t$, we quantify this difference by calculating the branch-specific loss for each sample $x_i$, and map the normalized loss difference to a probability score $p_i^{(t)} \in (0, 1)$ with the sigmoid function. This process can be expressed as:

$$p_i^{(t)} = \sigma\left(\mathcal{L}_{ce}(f(x_i; \theta^n), \tilde{y}_i) - \mathcal{L}_{ce}(f(x_i; \theta^c), \tilde{y}_i)\right), \quad (5)$$

where $\sigma(\cdot)$ denotes the sigmoid function. When $p \geq 0.5$, we consider it a clean sample and assign it to the clean branch for training, otherwise, it is assigned to the noisy branch for training. The decision threshold is set to $0.5$, which corresponds to equal losses for both branches under the sigmoid mapping.

While $p_i^{(t)}$ provides an immediate signal, relying solely on instantaneous predictions is prone to instability due to the stochastic nature of SGD. Direct utilization of raw probabilities can lead to oscillating assignments, where samples frequently flip between branches. To stabilize the training dynamics, we employ an EMA mechanism to accumulate historical predictions. Let $P_i^{(t)}$ denote the accumulated clean probability. The update rule is formulated as:

$$P_i^{(t)} = \lambda P_i^{(t-1)} + (1 - \lambda)p_i^{(t)}, \quad (6)$$

where $\lambda \in [0, 1)$ is the momentum coefficient. This temporal accumulation ensures clean probability reflects consistent historical behavior of the samples, avoiding instantaneous training noise.

**Probability-Guided Optimization.** Based on the accumulated clean probability $P_i^{(t)}$, we perform a dynamic partition of the local dataset into clean and noisy subsets:

$$
\begin{aligned}
\mathcal{D}_k^c &= \{(x_i, \tilde{y}_i) \mid P_i^{(t)} \geq 0.5\}, \\
\mathcal{D}_k^n &= \{(x_i, \tilde{y}_i) \mid P_i^{(t)} < 0.5\}.
\end{aligned}
\quad (7)
$$

The clean branch $\theta^c$ minimizes the risk on $\mathcal{D}_k^c$ to refine robust decision boundaries, where $A_k$ and $B_k^c$ are optimized simultaneously. The noisy branch $\theta^n$ minimizes the risk on $\mathcal{D}_k^n$ to learn noisy gradients. Crucially, during the optimization of the noisy branch, we freeze $A_k$ and only update $B_k^n$ to ensure that noise-specific variations are isolated within

the up-projection layer without distorting the shared feature subspace. This probability-guided mechanism ensures that clean and noisy data are stably separated as training progresses. The noisy branch $\theta_k^n$ serves as a noise estimator, facilitating noise detection through discrepancies between the two branches. The inference process uses only the clean branches $\theta_k^c$.

### 3.4. Noisy Branch Negative Learning

After the sample partitioning stabilizes in the initial decoupled LoRA fine-tuning stage, directly discarding the identified noisy samples incurs a significant information loss. To fully exploit the noisy subset $\mathcal{D}_k^n$, we propose a noisy branch negative learning method. This strategy employs the clean branch as a positive teacher to provide reliable soft probability targets for noisy samples, while leveraging the noisy branch as a negative teacher for negative learning to prevent noise overfitting (Kim et al., 2019).

**Clean Branch Positive Guidance.** For samples identified as noisy, their original labels are unreliable. Conversely, the clean branch $\theta_k^c$ captures robust feature representations, thus providing more reliable supervisory information. To leverage this, we utilize the prediction distribution of the clean branch as a positive guidance signal for the identified noisy subset. Specifically, we construct refined soft targets $\hat{y}_i^c = softmax(f(x_i; \theta_k^c))$ based on the clean-branch logits, which helps capture latent inter-class relationships while mitigating biases from noisy labels.

**Noise Branch Negative Guidance.** The clean branch focuses on robust semantic information, while the noise branch $\theta_k^n$ effectively captures noise patterns. We leverage this by treating $\theta_k^n$ as a negative teacher. We maximize the distributional difference between the two branches on noisy data to prevent the clean branch from learning noise patterns. Let $q_i^n = \mathrm{softmax}(f(x_i; \theta_k^n)/\tau)$ and $q_i^c = \mathrm{softmax}(f(x_i; \theta_k^c)/\tau)$ denote the predicted probability distributions of the noisy and clean branches, respectively. We stabilize the noisy-branch distribution by epsilon flooring and renormalization:

$$\tilde{q}_{i,j}^n = \frac{\max(q_{i,j}^n, \epsilon)}{\sum_{m=1}^C \max(q_{i,m}^n, \epsilon)}, \qquad (8)$$

where $\epsilon$ is a small constant for numerical stability and is set to 1e-6. Then, to prevent the clean branch from fitting the noise manifold, we minimize the negative Kullback-Leibler (KL) divergence, which effectively maximizes the distance between their predictive distributions:

$$\mathcal{L}_{kl}(x_i) = -KL(q_i^c \parallel \tilde{q}_i^n). \qquad (9)$$

This objective penalizes clean branch for producing a probability distribution similar to noisy branch, thereby pushing the clean decision boundary away from noise interference.

Finally, we integrate these components into a unified objective for local fine-tuning. The supervision targets are adaptive based on the sample clean probability. For clean samples $x_i \in \mathcal{D}_k^c$, we utilize the one-hot encoded original label $\mathbf{e}_{\tilde{y}_i}$. For noisy samples $x_i \in \mathcal{D}_k^n$, we utilize the refined soft target $\hat{y}_i^c$. The generalized cross-entropy (GCE) (Zhang & Sabuncu, 2018) loss is minimized over the entire local dataset. The GCE loss serves as a robust generalization between Mean Absolute Error (MAE) and CE. The total loss $\mathcal{L}_{\mathrm{total}}$ is formulated as:

$$\min_{\theta_k^c} \sum_{x_i \in D_k} \left[ \mathcal{L}_{gce}(x_i, y_i^\star) + \eta \cdot \mathbb{I}(x_i \in D_k^n) \cdot \mathcal{L}_{kl}(x_i) \right],$$
$$(10)$$

where $y_i^\star = \mathbf{e}_{\tilde{y}_i}$ if clean, and $y_i^\star = \hat{y}_i^c$ otherwise. The indicator function $\mathbb{I}(\cdot)$ ensures that negative learning is applied exclusively to the noisy subset, and $\eta$ is a hyperparameter balancing the impact of noise suppression.

### 3.5. Asymmetric Global Aggregation

Recent studies (Guo et al., 2025) show that the projection matrix $A$ exhibits high cosine similarity across clients, while the up-projection matrix $B$ varies substantially due to data heterogeneity. Given the structural stability of $A$ across clients, aggregating $A$ yields minimal federated benefits. Conversely, the high variance of the $B$ matrix reflects diverse local semantic distributions. Aggregating $B$ facilitates complementary knowledge exchange, and the diversity of local $B$ matrices enables the global model to integrate rich heterogeneous information. Thus, we only perform global aggregation on the up-projection matrices $B^c$ and $B^n$, while keeping $A_k$ local. This method achieves collaborative knowledge fusion across clients while alleviating federated aggregation bias.

At each round, the server aggregates the clean and noisy up-projection matrices separately, while keeping the down-projection matrix $A_k$ local. Specifically, the aggregation is formulated as:

$$B_g^c = \sum_{k=1}^K \frac{N_k}{N} B_k^c,$$
$$\qquad (11)$$
$$B_g^n = \sum_{k=1}^K \frac{N_k}{N} B_k^n.$$

In the next round, clients initialize their clean and noisy up-projection matrices with $B_g^c$ and $B_g^n$, respectively, while retaining their local $A_k$ for feature alignment. This separate aggregation prevents clean and noisy semantics from being mixed into a single global adapter. Since $B_k^c$ is optimized mainly on samples with high clean probability, aggregating $B^c$ builds a robust clean-branch consensus and facilitates knowledge sharing across heterogeneous clients. In contrast, $B_k^n$ is used to capture noise-specific patterns. Aggregating

*Table 1.* Performance comparison with SOTA federated LoRA methods on CIFAR-100. We report the average test accuracy (%) across all clients, per-round computation cost (s), and the trainable parameters (%) relative to FFT. The best results are highlighted in **bold**.

| Method | $\mu = 0.2$ | | $\mu = 0.4$ | | Computation Cost | Trainable Parameter |
|---|---|---|---|---|---|---|
| | Pairflip | Symflip | Pairflip | Symflip | | |
| FedLR (Zhang et al., 2023b) | 78.31 | 85.42 | 60.11 | 85.26 | 755.99 | 0.3503 |
| FFALoRA (Sun et al., 2024) | 78.52 | 83.91 | 57.86 | 81.34 | 506.36 | 0.5221 |
| RoLoRA (Chen et al., 2025) | 77.54 | 88.09 | 56.36 | 85.48 | 757.01 | 0.5221 |
| FLoRA (Wang et al., 2024) | 78.43 | 84.11 | 60.84 | 76.99 | 497.41 | 0.3503 |
| FlexLoRA (Bai et al., 2024) | 83.36 | 85.61 | 61.70 | 85.80 | 320.90 | 0.6075 |
| LoRA-FAIR (Bian et al., 2025) | 83.47 | 87.98 | 60.20 | 85.56 | 770.35 | 0.3503 |
| SLoRA (Babakniya et al., 2023) | 85.27 | 89.18 | 63.58 | 87.18 | 962.29 | 0.8038 |
| FedSA-LoRA (Guo et al., 2025) | 76.23 | 78.09 | 55.57 | 78.00 | 522.99 | 0.6075 |
| RFedLR (Fang & Ye, 2025) | 83.58 | 87.11 | 63.06 | 83.73 | 796.28 | 0.5221 |
| **FedDR-LoRA** | **91.43** | **91.37** | **88.30** | **90.92** | 646.93 | 0.4351 |

$B^n$ provides a global noisy branch, which improves subsequent sample separation and negative learning. The noisy branch is used only during training and is discarded during inference. Therefore, the final prediction relies solely on the clean branch, preventing the noise patterns captured by $B^n$ from being directly used for evaluation.

## 4. Experiments

### 4.1. Experimental Settings

**Models and Datasets.** Our FL setting employs a 12-layer Vision Transformer (ViT) model (Dosovitskiy et al., 2021) as the backbone for evaluation on vision tasks. The model is pre-trained on ImageNet-21K (Deng et al., 2009), processes input images of size 224×224, and divides them into 16×16 image patches. We use CIFAR-100 (Krizhevsky et al., 2009) as the benchmark dataset. It consists of 60,000 images spanning 100 classes. The dataset is distributed across $K = 10$ clients, with the data on each client being non-IID. We control the degree of non-IID by setting the concentration parameter $\beta$ of the Dirichlet distribution to 0.5.

**Label Noise Setup.** We adopt two common types of label noise. The Pair flip (Pairflip) (Han et al., 2018) noise replaces the label with a predefined similar class, while the Symmetric flip (Symflip) (Van Rooyen et al., 2015) noise randomly replaces the label with any other class. Different noise ratios of $\mu = 0.2, 0.4$ are applied to evaluate the robustness of the method.

**Comparison Methods.** We evaluate FedDR-LoRA against various SOTA methods, including federated LoRA variants and label noise learning algorithms. To ensure a fair comparison, all the above methods are evaluated under the same experimental setup. A detailed description of each baseline and its specific configuration is provided in Appendix C.1.

**Implementation Details.** Our experiments are conducted on $K = 10$ clients with $T = 40$ communication rounds, in-

cluding $T_0 = 10$ rounds for dual-branch warm-up, $T_1 = 10$ rounds for decoupled LoRA fine-tuning, and the remaining rounds for noisy branch negative learning. We use an SGD optimizer and set the LoRA rank $r = 4$. The complete hyperparameter settings are detailed in Appendix C.2.

### 4.2. Comparison Results

**Comparison with SOTA Federated LoRA Methods.** We evaluate the proposed FedDR-LoRA against SOTA federated PEFT approaches, including FedLR, FFALoRA, SLoRA, and others. The comparative results on the CIFAR-100 dataset across various noise configurations are presented in Table 1. FedDR-LoRA consistently achieves superior accuracy relative to all baseline methods across tested noise types and ratios. This performance advantage is particularly evident under the challenging Pairflip noise setting with $\mu = 0.4$, which represents a highly asymmetric noise distribution. While standard methods such as FFALoRA and RoLoRA exhibit significant performance degradation, yielding accuracies of 57.86% and 56.36% respectively, FedDR-LoRA maintains a robust accuracy of 88.30%. This substantial margin demonstrates the efficacy of our decoupled architecture in preventing global model collapse under severe noise conditions, a capability that existing methods lack when applied to the sensitive low-rank manifold.

**Efficiency Comparison with SOTA methods.** Table 1 summarizes the per-round computation cost and the proportion of trainable parameters relative to FFT. For multi-stage methods such as SLoRA and our FedDR-LoRA, we report the average metrics across all stages to provide a comprehensive evaluation. The results show that although FedDR-LoRA is not the most lightweight method, it achieves the best trade-off between efficiency and performance. Specifically, the computational cost of FedDR-LoRA is 646.93 seconds, which is higher than the efficient method FlexLoRA but significantly lower than SLoRA. Similarly, due to the introduction of a decoupled noise branch, our parameter

*Table 2.* Performance comparison with robust baselines integrating FedLR with LNL Methods.

| Method | $\mu = 0.2$ | | $\mu = 0.4$ | |
|---|---|---|---|---|
| | Pairflip | Symflip | Pairflip | Symflip |
| FedLR | 78.31 | 85.42 | 60.11 | 85.26 |
| +SCE | 86.86 | 89.93 | 66.79 | 89.04 |
| +Relabel | 78.82 | 75.31 | 56.29 | 75.76 |
| +DivideMix | 87.76 | 88.12 | 70.12 | 88.00 |
| **FedDR-LoRA** | **91.43** | **91.37** | **88.30** | **90.92** |

*Table 3.* Ablation study of each component. The average test accuracy (%) across all local models is demonstrated.

| Components | | $\mu = 0.2$ | | $\mu = 0.4$ | |
|---|---|---|---|---|---|
| DLF | NNL | Pairflip | Symflip | Pairflip | Symflip |
| | | 88.08 | 88.44 | 85.41 | 88.26 |
| ✓ | | 90.54 | 90.66 | 87.37 | 89.82 |
| ✓ | ✓ | **91.43** | **91.37** | **88.30** | **90.92** |

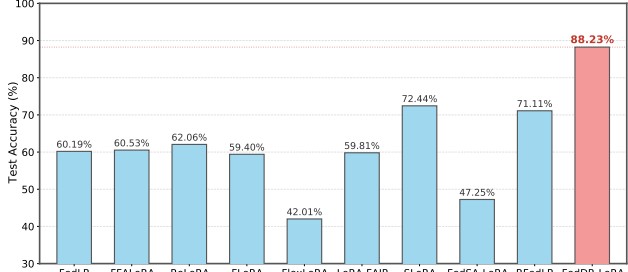

*Figure 3.* Performance comparison under instance-dependent noise with a noise rate of $\mu = 0.4$. We report the average test accuracy (%) across all local models.

count is slightly higher than the traditional FedLR. However, this small increase in resource consumption allows FedDR-LoRA to achieve the highest accuracy under all noise settings. These findings indicate that FedDR-LoRA effectively utilizes a moderate amount of extra parameters and computational resources, thereby significantly improving robustness.

**Comparison with Label Noise Learning Methods.** To verify that the performance gains stem from our specific architectural design rather than label noise learning algorithms, we compare FedDR-LoRA with strong baselines constructed by integrating FedLR with established LNL techniques, including SCE, Relabel, and DivideMix. These methods represent different common strategies for handling noisy labels, including robust loss design, pseudo-label correction, and sample division. As detailed in Table 2, although incorporating advanced LNL methods such as DivideMix improves the performance of FedLR to 70.12% under the Pairflip $\mu = 0.4$ setting, it remains significantly lower than the 88.30% achieved by FedDR-LoRA. This disparity indicates that standard LNL methods, which are typically designed for centralized training, are insufficient for federated environments constrained by low-rank adaptation. In contrast, FedDR-LoRA effectively disentangles noise from features by leveraging the functional asymmetry of the LoRA matrices, resulting in superior robustness.

### 4.3. Ablation Study

To rigorously investigate the individual contributions of the proposed components, we conduct an ablation study by incrementally integrating Decoupled LoRA Fine-tuning (DLF) and Noisy branch Negative Learning (NNL) into the baseline framework. The quantitative results are summarized in Table 3. To clarify, the first row in Table 3

corresponds to using only the dual-branch warm-up stage, without enabling DLF or NNL in subsequent training. We provide the results over three random seeds (0, 1, and 42) in Appendix D.3.

**Effectiveness of DLF.** The integration of decoupled LoRA fine-tuning significantly improves model performance under different noise types and noise rates. DLF utilizes the loss difference between the clean and noisy branches to estimate the clean probability of samples and assigns them to the corresponding branch for training. The noisy branch captures noise patterns, allowing the difference signal to effectively separate clean and noisy data, regardless of the noise type. The clean branch is trained on samples with high confidence, which protects the shared low-rank projection subspace and the clean branch from noise gradient interference. As shown in the second row of Table 3, DLF provides a significant performance improvement compared to the warm-up baseline. For example, under the setting of a noise rate $\mu = 0.2$ and noise type Pairflip, introducing DLF increases the accuracy from 88.08% to 90.54%.

**Effectiveness of NNL.** The subsequent incorporation of noisy branch negative learning further enhances model performance. As indicated in the third row of Table 3, the final system achieves accuracies of 91.43% and 88.30% under the Pairflip $\mu = 0.2$ and $\mu = 0.4$ settings, respectively. This incremental gain confirms the effectiveness of negative learning. Instead of discarding noisy samples, NNL utilizes them. It encourages the clean branch to diverge from the predictions of the noisy branch, mitigating overfitting to noise while preserving useful visual semantics. Therefore, compared to relying solely on clean samples filtered by DLF, the introduction of NNL better utilizes local data and achieves stronger generalization capabilities.

### 4.4. Robustness under Diverse Noise Scenarios

To further evaluate the robustness of FedDR-LoRA under more realistic and challenging noise distributions, we conduct experiments under instance-dependent noise and heterogeneous label noise distributions across clients.

*Table 4.* Performance comparison under heterogeneous label noise distributions across clients.

| Method | FedLR | FFALoRA | RoLoRA | FLoRA | FlexLoRA | LoRA-FAIR | SLoRA | FedSA-LoRA | RFedLR | FedDR-LoRA |
|---|---|---|---|---|---|---|---|---|---|---|
| Pairflip | 71.06 | 70.67 | 69.16 | 69.85 | 49.98 | 70.83 | 75.56 | 64.14 | 77.07 | **88.33** |
| Symflip | 83.09 | 79.22 | 83.64 | 80.87 | 53.02 | 83.14 | 86.46 | 70.92 | 85.97 | **90.95** |

*Table 5.* Performance comparison with SOTA federated LoRA methods on 20 Newsgroups.

| Method | $\mu = 0.2$ | | $\mu = 0.4$ | |
|---|---|---|---|---|
| | Pairflip | Symflip | Pairflip | Symflip |
| FedLR | 61.01 | 65.85 | 44.21 | 63.95 |
| FFALoRA | 59.69 | 64.88 | 43.24 | 63.21 |
| RoLoRA | 59.67 | 66.42 | 43.32 | 64.21 |
| FLoRA | 52.85 | 60.91 | 38.46 | 61.07 |
| FlexLoRA | 61.75 | 66.30 | 45.53 | 64.55 |
| LoRA-FAIR | 61.55 | 66.04 | 44.61 | 64.06 |
| SLoRA | 60.87 | 65.80 | 43.69 | 64.13 |
| FedSA-LoRA | 56.72 | 62.31 | 40.47 | 58.36 |
| RFedLR | 61.76 | 65.59 | 41.42 | 64.10 |
| **FedDR-LoRA** | **65.85** | **66.53** | **54.73** | **65.68** |

**Robustness against instance-dependent noise.** To evaluate the sensitivity of FedDR-LoRA to more realistic and complex noise distributions, we conduct experiments with instance-dependent noise. Instance-dependent noise corrupts labels according to input features, making ambiguous samples more likely to be mislabeled. Unlike class-dependent noise, its distribution varies across individual samples. The noise generation follows the methodology from (Xia et al., 2020). The overall noise rate is set to 0.4, and the models are evaluated over 30 communication rounds. The results (Figure 3) demonstrate that instance-dependent noise poses a severe challenge to existing baselines. However, FedDR-LoRA achieves an accuracy of 88.23%, outperforming the strongest baseline algorithms.

**Robustness under heterogeneous client noise rates.** We further consider a heterogeneous noise setting where different clients suffer from different noise rates. Following (Xu et al., 2022), we assign noise rates of $\{0.1, 0.2, 0.3, 0.4, 0.5\}$ across 10 clients, with two clients assigned to each rate. This setting introduces both data and noise heterogeneity, requiring the model to handle clients with different levels of label noise. Table 4 shows that FedDR-LoRA consistently outperforms all baselines when clients have heterogeneous noise rates. Under Pairflip noise, FedDR-LoRA improves the accuracy from 77.07% achieved by the strongest baseline RFedLR to 88.33%. Under Symflip noise, FedDR-LoRA improves the best baseline result from 86.46% to 90.95%. These results indicate that FedDR-LoRA can adapt to different noise severities across clients, since the noisy branch $B^n$ captures noise-specific patterns while the shared feature subspace maintained by $A$ remains relatively robust.

### 4.5. Effectiveness on NLP Tasks

To verify that our approach is not restricted to visual domains and retains strong generalization capability when transferred to other scenarios, we performed rigorous comparative experiments on Natural Language Processing (NLP) tasks, systematically assessing its adaptability and robustness across modalities. We adopt the 20 Newsgroups (20ng) (Lang, 1995) dataset as the benchmark for comparison, and perform experiments under the same noise settings as those used in the visual tasks. The training process involved $K = 3$ clients, and the RoBERTa-base model (Liu et al., 2019) is employed as the backbone. For text processing, we use a maximum sequence length of 128 tokens. Unlike the vision tasks, the NLP experiments utilize the AdamW (Loshchilov & Hutter, 2019) optimizer with a learning rate of $2 \times 10^{-4}$.

As shown in Table 5, FedDR-LoRA also significantly outperforms SOTA methods on the NLP task. Consistent with the visual tasks, this advantage is particularly pronounced under higher noise levels and the Pairflip noise setting. When the noise rate is 0.4 and the noise type is Pairflip, FedDR-LoRA demonstrates remarkable effectiveness, achieving an accuracy of up to 54.73%. This is 9.20% higher than the accuracy of the suboptimal method, FlexLoRA, indicating a significant performance advantage. These results confirm the strong robustness of FedDR-LoRA across different modalities and its resistance to extreme noise.

## 5. Conclusion

This paper investigates robust federated fine-tuning with LoRA under noisy labels. We reveal a robustness asymmetry in LoRA factorization. The down-projection matrix $A$ tends to learn robust and transferable representations, whereas the up-projection matrix $B$ is more susceptible to fitting corrupted supervision, thereby aggravating local overfitting and federated aggregation bias. Motivated by this observation, we propose FedDR-LoRA that decouples robust feature learning from noise modeling through a decoupled LoRA design, recovers semantic information from noisy samples via noisy branch negative learning, and adopts asymmetric aggregation by keeping $A$ local while aggregating $B$ to establish global consensus. Extensive experiments across diverse noisy federated scenarios demonstrate that FedDR-LoRA consistently outperforms SOTA baselines in both robustness and generalization.

## Acknowledgements

This work was supported by the National Natural Science Foundation of China under Grant (62361166629, T2541022), the National Key Research and Development Program of China under Grant (2026YFE0202100), and the Major Project of Science and Technology Innovation of Hubei Province under Grant (2025BEA002). The numerical calculations were supported by the supercomputing system at the Supercomputing Center of Wuhan University.

## Impact Statement

This paper presents work whose goal is to advance the field of machine learning. There are many potential societal consequences of our work, none of which we feel must be specifically highlighted here.

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

## A. The Algorithm of FedDR-LoRA

The pseudocode of FedDR-LoRA is shown in Algorithm 1.

---

**Algorithm 1** Training process of FedDR-LoRA

---

**Input:** Clients $K$, local datasets $\mathcal{D}_k$, frozen backbone $W_0$, LoRA rank $r$, scaling $\alpha$, total communication rounds $T$, hyperparameters $\lambda, \eta, \tau$.

1 **Server Execute:**
2 **for** *round* $t = 1, \dots, T$ **do**
3      **for** *each client* $k \in \{1, \dots, K\}$ **in parallel do**
4          Send global $(B_g^c, B_g^n)$ to client $k$
5          $(B_k^c, B_k^n) \leftarrow LocalUpdate(B_g^c, B_g^n, t)$
6      **end**
7      Separately aggregate the clean and noisy branches by Eq. (11) to update $(B_g^c, B_g^n)$
8 **end**

9 **Local Update:**
10 **Function** *LocalUpdate($B_g^c, B_g^n, t$)*
11      Receive $(B_g^c, B_g^n)$ from server and set local matrices $B_k^c \leftarrow B_g^c, B_k^n \leftarrow B_g^n$
12      Initialize local adapter $\theta_k = \{A_k, B_k^c, B_k^n\}$
13      **if** $t \leq T_0$ **then**
         // Dual-Branch LoRA Warm-up
14          Construct clean and noisy subsets by Eq. (3)
15          Update $\{A_k, B_k^c\}$ on $\mathcal{D}_k^c$ and update $B_k^n$ on $\mathcal{D}_k^n$
16      **else if** $t \leq T_0 + T_1$ **then**
         // Decoupled LoRA Fine-Tuning
17          Compute clean probability by Eq. (5) & (6)
18          Partition samples into $(\mathcal{D}_k^c, \mathcal{D}_k^n)$ by Eq. (7)
19          Update clean branch $\{A_k, B_k^c\}$ on $\mathcal{D}_k^c$
20          Update noisy branch $B_k^n$ on $\mathcal{D}_k^n$
21      **else**
         // Noisy-Branch Negative Learning
22          Construct soft targets for $\mathcal{D}_k^n$ with clean branch
23          Update $\theta_k^c$ with local dataset $D_k$ via Eq. (10)
24      **end**
25      **return** $(B_k^c, B_k^n)$ to server

---

## B. Robustness Asymmetry in LoRA Factors

### B.1. Theoretical Analysis

Our analysis provides a mechanism-level explanation of how label-noise perturbations propagate differently through the two LoRA factors in $\Delta W = BA$. Consider a linear

layer parameterized by a frozen pre-trained weight $W_0$ and a LoRA update $\Delta W = \frac{\alpha}{r} BA$, *i.e.*,

$$h = W_0 x + \frac{\alpha}{r} BAx, \qquad (12)$$

where $W_0$ is frozen, $A \in \mathbb{R}^{r \times d_{in}}$, $B \in \mathbb{R}^{d_{out} \times r}$, and $r \ll \min(d_{in}, d_{out})$. Let $\ell(h, \tilde{y})$ be any differentiable loss with a potentially noisy label $\tilde{y}$.

Define the backpropagated error signal at the layer output as $g \triangleq \nabla_h \ell(h, \tilde{y}) \in \mathbb{R}^{d_{out}}$, *i.e.*, the gradient of the loss with respect to the pre-activation $h$ under the noisy label $\tilde{y}$. Moreover, let $z \triangleq Ax \in \mathbb{R}^r$ denote the low-dimensional representation produced by the LoRA down-projection $A$, which serves as the rank-$r$ feature vector used by the up-projection $B$ to form the update $\Delta W x$.

**Lemma B.1** (Gradient asymmetry of LoRA factors). *For differentiable loss $\ell$, the gradients w.r.t. LoRA factors satisfy*

$$\begin{aligned} \nabla_B \ell &= \frac{\alpha}{r} g z^\top, \\ \nabla_A \ell &= \frac{\alpha}{r} B^\top g x^\top. \end{aligned} \qquad (13)$$

*Proof.* By the chain rule, the LoRA term contributes $\frac{\alpha}{r} BAx$ to the layer output $h$.

For $B$, note that

$$h = W_0 x + \frac{\alpha}{r} B(Ax) \quad \Rightarrow \quad dh = \frac{\alpha}{r} dB (Ax).$$

Therefore,

$$\begin{aligned} d\ell &= \langle \nabla_h \ell, dh \rangle \\ &= \frac{\alpha}{r} \langle g, dB (Ax) \rangle \\ &= \frac{\alpha}{r} \langle g(Ax)^\top, dB \rangle \\ &= \frac{\alpha}{r} \langle g z^\top, dB \rangle, \end{aligned}$$

which implies

$$\nabla_B \ell = \frac{\alpha}{r} g z^\top.$$

For $A$, note that

$$h = W_0 x + \frac{\alpha}{r} (BA)x \quad \Rightarrow \quad dh = \frac{\alpha}{r} B \, dA \, x.$$

Therefore,

$$\begin{aligned} d\ell &= \langle \nabla_h \ell, dh \rangle \\ &= \frac{\alpha}{r} \langle g, B \, dA \, x \rangle \\ &= \frac{\alpha}{r} \langle B^\top g, dA \, x \rangle \\ &= \frac{\alpha}{r} \langle (B^\top g) x^\top, dA \rangle, \end{aligned}$$

which implies

$$\nabla_A \ell = \frac{\alpha}{r} B^\top g \, x^\top.$$

$\square$

Table 6. Layer-wise cosine similarity between clean and noisy LoRA matrices for ViT on CIFAR-100.

| | Layer1 | Layer2 | Layer3 | Layer4 | Layer5 | Layer6 | Layer7 | Layer8 | Layer9 | Layer10 | Layer11 | Layer12 | Avg |
|---|---|---|---|---|---|---|---|---|---|---|---|---|---|
| Query $A$ | 0.9999 | 0.9999 | 1.0000 | 0.9988 | 0.9998 | 0.9999 | 0.9998 | 0.9998 | 0.9999 | 0.9998 | 0.9994 | 0.9986 | 0.9996 |
| Query $B$ | 0.4081 | 0.6725 | 0.5785 | 0.8766 | 0.8682 | 0.8165 | 0.7742 | 0.7505 | 0.7683 | 0.6431 | 0.8751 | 0.9183 | 0.7458 |
| Value $A$ | 0.9899 | 0.9851 | 0.9827 | 0.9877 | 0.9859 | 0.9894 | 0.9909 | 0.9921 | 0.9991 | 0.9993 | 0.9995 | 0.9975 | 0.9916 |
| Value $B$ | 0.7194 | 0.7830 | 0.8035 | 0.8485 | 0.8676 | 0.8340 | 0.8201 | 0.8072 | 0.7673 | 0.8177 | 0.7054 | 0.8683 | 0.8035 |

Table 7. Layer-wise cosine similarity between clean and noisy LoRA matrices for RoBERTa on 20 Newsgroups.

| | Layer1 | Layer2 | Layer3 | Layer4 | Layer5 | Layer6 | Layer7 | Layer8 | Layer9 | Layer10 | Layer11 | Layer12 | Avg |
|---|---|---|---|---|---|---|---|---|---|---|---|---|---|
| Query $A$ | 0.9825 | 0.9854 | 0.9820 | 0.9843 | 0.9818 | 0.9767 | 0.9644 | 0.9705 | 0.9555 | 0.9736 | 0.9522 | 0.9659 | 0.9729 |
| Query $B$ | 0.1402 | 0.2510 | 0.2260 | 0.3871 | 0.5097 | 0.4483 | 0.6945 | 0.5466 | 0.5051 | 0.6463 | 0.6193 | 0.7166 | 0.4742 |
| Value $A$ | 0.9870 | 0.9860 | 0.9845 | 0.9829 | 0.9863 | 0.9852 | 0.9883 | 0.9786 | 0.9837 | 0.9841 | 0.9842 | 0.9730 | 0.9837 |
| Value $B$ | 0.4666 | 0.5141 | 0.5307 | 0.6052 | 0.6994 | 0.7634 | 0.7201 | 0.5239 | 0.6876 | 0.6638 | 0.6801 | 0.6542 | 0.6258 |

Table 8. Parameter analysis for the smoothing factor $\lambda$ in EMA. The analysis is conducted under the 40% Symflip noise setting.

| $\lambda$ | 0.5 | 0.6 | 0.7 | 0.8 | 0.9 |
|---|---|---|---|---|---|
| Acc | 90.87 | 90.86 | 90.90 | 90.92 | 90.91 |

Table 9. Parameter analysis for the temperature parameter $\tau$. The analysis is conducted under the 40% Symflip noise setting.

| $\tau$ | 0.5 | 0.6 | 0.7 | 0.8 | 0.9 |
|---|---|---|---|---|---|
| Acc | 90.81 | 90.82 | 90.91 | 90.89 | 90.92 |

Table 10. Parameter analysis for the negative learning weight $\eta$ under 40% Symflip noise.

| $\eta$ | 0.1 | 0.2 | 0.3 | 0.4 | 0.5 |
|---|---|---|---|---|---|
| Acc | 90.86 | 90.88 | 90.92 | 90.92 | 90.90 |

Table 11. Parameter analysis for the negative learning weight $\eta$ under 40% Pairflip noise.

| $\eta$ | 0.01 | 0.04 | 0.07 | 0.1 |
|---|---|---|---|---|
| Acc | 88.30 | 87.14 | 86.70 | 84.43 |

*Remark* B.2 (Asymmetric noise propagation in LoRA factors). Let $g_y = \nabla_h \ell(h, y)$ and $g_{\tilde{y}} = \nabla_h \ell(h, \tilde{y})$ denote the output-side error signals under the clean label $y$ and the noisy label $\tilde{y}$, respectively, and define the noise-induced perturbation $\delta g \triangleq g_{\tilde{y}} - g_y$.

From Lemma B.1, the corresponding gradient perturbations on the LoRA factors are

$$\delta(\nabla_B \ell) = \frac{\alpha}{r} \delta g \, (Ax)^\top,$$
$$\delta(\nabla_A \ell) = \frac{\alpha}{r} B^\top \delta g \, x^\top.$$

This exhibits a structural asymmetry, *i.e.*, label noise affects $B$ through the raw error perturbation $\delta g$, whereas it affects $A$ through the projected perturbation $B^\top \delta g$.

To quantify this asymmetry, we upper bound the magnitude of the perturbations using standard norm inequalities for outer products and operator norms:

$$\left\| \delta(\nabla_B \ell) \right\|_F = \frac{\alpha}{r} \left\| \delta g \, (Ax)^\top \right\|_F$$
$$= \frac{\alpha}{r} \left\| \delta g \right\|_2 \left\| Ax \right\|_2,$$
$$\left\| \delta(\nabla_A \ell) \right\|_F = \frac{\alpha}{r} \left\| B^\top \delta g \, x^\top \right\|_F$$
$$= \frac{\alpha}{r} \left\| B^\top \delta g \right\|_2 \left\| x \right\|_2$$
$$\leq \frac{\alpha}{r} \left\| B \right\|_2 \left\| \delta g \right\|_2 \left\| x \right\|_2.$$

Therefore, the perturbation to $\nabla_A \ell$ caused by noise is controlled by the linear operator $B^\top$. Specifically, its magnitude is upper bounded by $\|B\|_2 \|\delta g\|_2 \|x\|_2$ up to the constant factor $\alpha/r$. In contrast, the perturbation to $\nabla_B \ell$ depends on

the original error perturbation $\delta g$ and is not modulated by other operators. Consequently, the noise only affects $A$ after being projected through $B^\top$, so its influence is limited to the rank-$r$ subspace generated by the LoRA update.

The noise can influence $A$ only through the projected signal $B^\top \delta g$, so only the component of $\delta g$ that lies in the column space of $B$ contributes to updating $A$. Equivalently, decompose the noise-induced error perturbation into components parallel and orthogonal to the column space of $B$,

$$\delta g = \delta g_\| + \delta g_\perp$$

with $\delta g_\| \in \mathrm{col}(B)$ and $\delta g_\perp \perp \mathrm{col}(B)$. Then

$$B^\top \delta g = B^\top (\delta g_\| + \delta g_\perp)$$
$$= B^\top \delta g_\| + B^\top \delta g_\perp.$$

Since $\delta g_\perp \perp \mathrm{col}(B)$ and $Bv \in \mathrm{col}(B)$ for any $v$, we have

$$(B^\top \delta g_\perp)^\top v = \langle \delta g_\perp, Bv \rangle = 0, \quad \forall v,$$

which implies $B^\top \delta g_\perp = 0$, and therefore

$$B^\top \delta g = B^\top \delta g_\|.$$

Thus, the influence of noise on $A$ is restricted to a subspace of rank at most $r$, which provides a structural explanation for why $A$ remains empirically stable even during convergence.

### B.2. Empirical Validation

To further validate the robustness asymmetry between the two LoRA factors, we provide additional layer-wise cosine similarity analysis between LoRA matrices trained on clean

*Table 12.* Rank sensitivity analysis under different LoRA ranks $r$, where the scaling factor $\alpha$ is adjusted proportionally with $r$. We report the average test accuracy (%) under 40% Symflip noise.

| Rank | 2 | 4 | 8 | 16 | 32 |
|---|---|---|---|---|---|
| FedLR | 86.39 | 85.26 | 86.47 | 85.43 | 85.85 |
| FFALoRA | 79.61 | 81.34 | 82.45 | 82.49 | 83.28 |
| RoLoRA | 85.19 | 85.48 | 85.71 | 85.86 | 86.12 |
| FLoRA | 77.45 | 76.99 | 78.07 | 79.25 | 80.24 |
| FlexLoRA | 85.46 | 85.80 | 85.05 | 86.68 | 86.19 |
| LoRA-FAIR | 85.75 | 85.56 | 85.00 | 84.67 | 84.90 |
| SLoRA | 86.31 | 87.18 | 87.63 | 88.16 | 88.46 |
| FedSA-LoRA | 78.82 | 78.00 | 78.19 | 75.74 | 75.73 |
| RFedLR | 82.48 | 83.73 | 82.86 | 84.38 | 83.51 |
| **FedDR-LoRA** | **90.80** | **90.92** | **90.98** | **91.10** | **90.85** |

and noisy datasets. To compute the similarity, we initialize two identical LoRA-based models and fine-tune them separately on clean and noisy datasets. After training, we extract the down-projection matrix A and up-projection matrix B from the Query and Value modules across all layers of both models. These weight matrices are then flattened into one-dimensional vectors. Finally, we calculate the cosine similarity between the corresponding flattened vectors of the clean and noisy models to quantitatively evaluate the parameter deviation caused by label noise. Figure 1 visualizes this phenomenon on a ViT-based LoRA model trained on CIFAR-100. Here, we provide the detailed numerical results and further extend the analysis to a RoBERTa-based LoRA model on the 20 Newsgroups dataset.

As shown in Tables 6 and 7, the down-projection matrix $A$ consistently maintains high cosine similarity between clean and noisy training settings across both visual and textual tasks. In contrast, the up-projection matrix $B$ exhibits much lower similarity, indicating that it changes more substantially to fit corrupted supervision. These results provide broader empirical evidence that $A$ is relatively stable under label noise, whereas $B$ is more sensitive to noisy targets. This observation supports the decoupled design of FedDR-LoRA, where the stable feature subspace in $A$ is preserved while the noise-sensitive behavior of $B$ is explicitly modeled through separate clean and noisy branches.

## C. Experimental Details

### C.1. Comparison Methods

We compare FedDR-LoRA with a series of SOTA methods in the field under noisy label settings. FedLR is a standard federated fine-tuning method using LoRA. FFA-LoRA (Sun et al., 2024) freezes matrix $A$ and updates only matrix $B$. RoLoRA (Chen et al., 2025) alternately freezes the LoRA $A$ and $B$ matrices. FLoRA (Wang et al., 2024) proposes a stacking-based aggregation mechanism that is applicable to heterogeneous LoRA ranks among clients. FlexLoRA

(Bai et al., 2024) employs singular value decomposition to allow LoRA matrices of different ranks to be combined. LoRA-FAIR (Bian et al., 2025) introduces a correction matrix $B$ to address parameter aggregation inconsistency. SLoRA (Babakniya et al., 2023) obtains cumulative weight updates through sparse fine-tuning, and then applies singular value decomposition to construct low-rank factors, thereby initializing the LoRA adapter. FedSA-LoRA (Guo et al., 2025) transmits only matrix $A$ to the server for aggregation, while keeping matrix $B$ local. RFedLR (Fang & Ye, 2025) is a robust federated PEFT framework that enhances local robustness through sensitivity-aware robust tuning and mitigates noise propagation through adaptive federated LoRA aggregation. To further strengthen the comparison, we include a FedLR baseline equipped with SOTA Label Noise Learning (LNL) methods. SCE (Wang et al., 2019) introduces a noise-robust loss function. DivideMix (Li et al., 2020a) adapts methods from semi-supervised learning. Tanaka *et al.* (Tanaka et al., 2018) propose a noise-learning method that includes a Relabel strategy. To ensure a fair comparison, all the above methods are evaluated under the same experimental setup.

### C.2. Implementation Details

In our experiments, we set the number of clients to $K = 10$, with a total of $T = 40$ training rounds, including $T_0 = 10$ rounds for the dual-branch warm-up phase, $T_1 = 10$ rounds for the decoupled LoRA fine-tuning phase, and the remaining rounds for noisy branch negative learning. After each round, server-side parameter aggregation is performed. We use an SGD optimizer with a learning rate of 0.01, momentum of 0.9, and weight decay of $1e - 4$. The batch size for the training set is set to 128. The rank $r$ of LoRA is set to 4, and the scaling factor $\alpha$ is also set to 4. The loss threshold $\delta_{low}$ is a dynamic quantile threshold. During the warm-up stage, $\delta_{low}$ starts high to include most samples, then anneals to focus on clean ones. Specifically, $\delta_{low}$ is set as a quantile threshold of the per-sample losses from the clean branch, whose quantile decreases linearly across epochs from the 100th percentile to the 10th percentile, thereby progressively selecting lower-loss samples. In contrast, $\delta_{high}$ is fixed at the 90th percentile of the loss distribution, consistently selecting the top 10% highest-loss samples as the noisy subset. The smoothing factor $\lambda$ in the EMA stage is 0.8. The hyperparameter used to balance CE and MAE in the GCE loss calculation is set to 0.9. For the negative learning phase, the temperature $\tau$ is set to 0.9. The weight of the KL divergence loss $\eta$ is specifically configured as 0.01 for Pairflip noise and 0.3 for Symflip noise. Symflip noise is uniformly distributed across all classes, requiring a stronger penalty $\eta = 0.3$ to ensure effective branch divergence. In contrast, Pairflip noise is concentrated in specific class pairs, forming a structured perturbation. The clean branch is sensitive to

*Table 13.* Ablation study of each component. The average test accuracy (%) across all local models is demonstrated.

| Components | | $\mu = 0.2$ | | $\mu = 0.4$ | |
|---|---|---|---|---|---|
| DLF | NNL | Pairflip | Symflip | Pairflip | Symflip |
| | | 88.22±0.14 | 88.53±0.09 | 84.99±0.76 | 88.56±0.28 |
| ✓ | | 90.47±0.14 | 90.63±0.06 | 88.00±1.10 | 89.94±0.16 |
| ✓ | ✓ | **91.43±0.09** | **91.15±0.19** | **88.93±1.03** | **90.98±0.25** |

*Table 14.* Performance comparison on CIFAR-100 in a large-scale federated setting with 100 total clients and 10% participation rate per round under 40% label noise. We report the average test accuracy (%) of the local models.

| Method | FedLR | FFALoRA | RoLoRA | FLoRA | FlexLoRA | LoRA-FAIR | SLoRA | FedSA-LoRA | RFedLR | FedDR-LoRA |
|---|---|---|---|---|---|---|---|---|---|---|
| Pairflip | 45.39 | 42.61 | 43.09 | 45.28 | 14.62 | 43.95 | 36.92 | 42.62 | 46.62 | **47.75** |
| Symflip | 64.45 | 63.68 | 65.18 | 66.22 | 19.88 | 65.02 | 54.67 | 63.66 | 67.36 | **69.72** |

this concentrated error, thus a smaller weight $\eta = 0.01$ is utilized to prevent over-suppression.

## D. Additional Experimental Results

### D.1. Hyperparameter Sensitivity Analysis

We conduct comprehensive ablation studies on the exponential moving average decay rate $\lambda$, temperature $\tau$, and negative learning weight $\eta$. For the smoothing factor $\lambda$ in EMA, the performance remains remarkably stable between 0.5 and 0.9. The results (Table 8) demonstrates that the framework is highly robust to $\lambda$. Evaluating the temperature parameter $\tau$ from 0.5 to 0.9 yields consistent accuracy around 90.8% (Table 9). The method maintains optimal performance across different temperature scales. Regarding the negative learning weight $\eta$ under Symflip noise, the accuracy remains stable across values from 0.1 to 0.5 (Table 10). Under the Pairflip noise, setting $\eta$ to 0.01 yields the best accuracy of 88.30% (Table 11). Pairflip noise is concentrated in specific class pairs, forming a structured perturbation. The clean branch is sensitive to this concentrated error, thus a smaller weight $\eta = 0.01$ is utilized to prevent over-suppression.

### D.2. Sensitivity Analysis to LoRA Rank

To evaluate the sensitivity of FedDR-LoRA to the LoRA rank, we conduct experiments with different ranks $r \in \{2, 4, 8, 16, 32\}$ under 40% Symflip label noise. Following the common LoRA setting, the scaling factor $\alpha$ is adjusted proportionally with the rank to maintain a comparable update magnitude. As shown in Table 12, FedDR-LoRA consistently outperforms all baseline methods across different LoRA ranks. Moreover, its performance remains stable as the rank varies from 2 to 32. This suggests that the robustness of FedDR-LoRA does not mainly come from increasing the adapter capacity, but from separating robust feature learning from noise modeling. Once the rank is sufficient to capture the task-relevant low-rank subspace, increasing

the rank further brings only marginal gains, leading to a relatively flat performance trend.

### D.3. Results over Multiple Random Seeds

To further evaluate the stability of FedDR-LoRA, we repeat the ablation experiments using three different random seeds, i.e., 0, 1, and 42. The ablation study results (Mean ± Standard Deviation) are summarized in Table 13. The results show that both DLF and NNL consistently improve the performance under different noise settings, demonstrating the robustness and stability of each component.

### D.4. Performance in Large-Scale Scenarios

We evaluate FedDR-LoRA on a larger-scale federated setting with $K = 100$ total clients, where 10 clients are sampled in each communication round, and the concentration parameter $beta$ of the Dirichlet distribution controlling the degree of Non-IID is set to 0.5. In a FL scenario with 100 clients, each client has much less local data, the non-IID level is higher, and only 10% of clients participate in each round. Under such a setting, all methods suffer clear performance degradation. In this context, FedDR-LoRA still achieves the best performance under both Pairflip and Symflip noise. As shown in Table 14, FedDR-LoRA achieves an accuracy of 47.75% with Pairflip noise and 69.72% with Symflip noise, outperforming the strongest baseline algorithm, RFedLR. The result suggests that the advantage of FedDR-LoRA is not restricted to the small-scale full-participation setting.

In the FedDR-LoRA implementation, some clients may not be selected during the first $T_0 + T_1$ rounds and thus have not yet established discrepancy-based clean probabilities. These clients first perform one round of discrepancy-based probability estimation before noisy-branch negative learning. This ensures that the clean and noisy partition is available before applying NNL, and we find that the mechanism remains effective with 100 clients.

*Table 15.* Performance comparison under a severe data heterogeneity setting with Dirichlet concentration parameter $\beta = 0.1$ and 40% label noise. We report the average test accuracy (%) across all local models.

| Method | FedLR | FFALoRA | RoLoRA | FLoRA | FlexLoRA | LoRA-FAIR | SLoRA | FedSA-LoRA | RFedLR | FedDR-LoRA |
|---|---|---|---|---|---|---|---|---|---|---|
| Pairflip | 56.24 | 52.77 | 56.08 | 54.68 | 33.79 | 55.77 | 58.76 | 47.25 | 57.15 | **86.78** |
| Symflip | 74.86 | 61.28 | 74.40 | 63.50 | 33.54 | 75.36 | 80.07 | 53.46 | 72.99 | **89.81** |

*Table 16.* Performance comparison on Tiny-ImageNet under 40% noise. We report the average test accuracy (%) across all local models.

| Method | FedLR | FFALoRA | RoLoRA | FLoRA | FlexLoRA | LoRA-FAIR | SLoRA | FedSA-LoRA | RFedLR | FedDR-LoRA |
|---|---|---|---|---|---|---|---|---|---|---|
| Pairflip | 61.81 | 59.77 | 61.78 | 62.68 | 50.17 | 63.44 | 65.29 | 57.99 | 62.68 | **87.07** |
| Symflip | 87.33 | 83.81 | 87.25 | 86.84 | 71.73 | 87.50 | 89.08 | 78.89 | 88.09 | **90.64** |

### D.5. Effectiveness under Severe Data Heterogeneity

To further demonstrate the robustness of FedDR-LoRA under severe data heterogeneity, we conduct a supplementary evaluation under an extreme Non-IID setting, where the Dirichlet concentration parameter is set to $\beta = 0.1$ and noise rate is 0.4. The results are reported in Table 15. As shown in Table 15, FedDR-LoRA consistently achieves the best performance under severe data heterogeneity. In particular, under Pairflip noise, FedDR-LoRA improves the accuracy from 58.76% achieved by the strongest baseline SLoRA to 86.78%. Under Symflip noise, FedDR-LoRA also outperforms all baselines, improving the accuracy from 80.07% to 89.81%. These results indicate that FedDR-LoRA remains robust when label noise is coupled with strong client distribution shifts.

### D.6. Effectiveness on Tiny-ImageNet

To further validate the generalization capability of FedDR-LoRA on more complex visual tasks, we conduct supplementary experiments on the Tiny-ImageNet dataset. Tiny-ImageNet is a subset of ImageNet, containing 200 classes with 500 training images per class at a resolution of $64 \times 64$. We set the noise rate to $\mu = 0.4$ and keep the other experimental settings consistent with the main experiments. The results are reported in Table 16.

As shown in Table 16, FedDR-LoRA consistently outperforms all baseline methods on Tiny-ImageNet. The advantage is particularly pronounced under Pairflip noise, where FedDR-LoRA achieves 87.07% accuracy, outperforming the strongest baseline SLoRA by 21.78%. This result demonstrates that FedDR-LoRA remains effective on more challenging visual recognition tasks with a larger number of classes. Under Symflip noise, FedDR-LoRA also achieves the best performance, improving the accuracy from 89.08% to 90.64%. These results further confirm that the robustness of FedDR-LoRA is not limited to CIFAR-100, but can generalize to more complex visual datasets.

