# OpenReview forum: "Decoupled Low-Rank Adaptation for Robust Federated Fine-Tuning"
_ICML.cc/2026/Conference — ICML 2026 regular_

### Official Review · Reviewer_M1Ek · 2026-02-26

**Soundness:** 2
**Presentation:** 2
**Significance:** 2
**Originality:** 2
**Overall Recommendation:** 4
**Confidence:** 4

**Summary:**

This paper studies robust federated fine-tuning of pre-trained models using LoRA under label noise. The authors observe an empirical asymmetry between the LoRA matrix A and matrix B, arguing that A is relatively robust to label noise while B is more susceptible to corrupted supervision. Based on this observation, the paper proposes FDLoRA, which introduces a dual-branch LoRA design with separate clean and noisy branches, dynamic sample routing via discrepancy-based probability estimation, noisy-branch negative learning, and asymmetric aggregation that aggregates only the B matrices while keeping A local.

**Compliance With Llm Reviewing Policy:**

Affirmed.

**Final Justification:**

During the rebuttal, the authors addressed most of my concerns. Thus, I determined to increase my score to 4.

**Key Questions For Authors:**

1.	If the LoRA matrix A is intrinsically robust to label noise as claimed, why is it necessary to freeze A in the noisy branch?
2.	How does the proposed dual-branch design behave under heterogeneous noise distributions across clients (e.g., varying noise rates per client or client-specific corruption patterns)?
3.	Given the additional complexity introduced by dual branches and multi-stage training, what is the communication and memory overhead compared to simpler federated LoRA baselines?

**Limitations:**

yes

**Strengths And Weaknesses:**

Strengths:

1.	The topic of federated fine-tuning of foundation models is timely and robust federated fine-tuning under label noise has been limited studied.
2.	The reported performance improvements under severe noise (e.g., high pair-flip rates) are substantial, suggesting the method is effective in label noise setting.
3.	The inclusion of both vision and NLP classification tasks provides some evidence of generality beyond a single domain.

Weaknesses:

1.	The central motivation claims that the LoRA A is intrinsically robust to label noise. However, in the proposed dual-branch design, the noisy branch explicitly freezes A to prevent corruption. This makes me feel the key story ("A is robust") conditional on additional training heuristics rather than a truly structural property.
2.	The main insight about the asymmetry between A and B is not particularly novel. Prior LoRA literature (in both centralized [1] and FL [2][3]) has repeatedly observed that A tends to capture more general/shared representations (less label-dependent), while B is more sensitive to labels and often behaves more “personalized.”
3.	The overall training pipeline is quite complex (warm-up + dynamic routing via EMA + negative learning + dual branches), but the method feels only loosely tied to FL. Much of the design resembles a centralized noisy-label learning recipe, and the paper does not convincingly explain what FL-specific challenges (e.g., client drift, heterogeneous noise across clients) are being uniquely addressed.
4.	The only part specifically designed for FL is Section 3.5 Asymmetric Global Aggregation. But this just resembles previously explored selective aggregation paradigms from FedSA [2].
5.	The evaluation focuses exclusively on classification tasks with synthetic label noise. It is unclear if the proposed approach can extend to generative LLM fine-tuning scenarios which I believe should be more important in LLM era.
6.	The experiments lack sensitivity studies on key federated and LoRA hyperparameters, including number of clients, LoRA rank, local epochs, and noise heterogeneity across clients. Furthermore, results are reported without variance across multiple random seeds.

[1] HydraLoRA: An Asymmetric LoRA Architecture for Efficient Fine-Tuning. NeurIPS 2024

[2] Selective Aggregation for Low-Rank Adaptation in Federated Learning. ICLR 2025

[3] Adaptive LoRA Experts Allocation and Selection for Federated Fine-Tuning. NeurIPS 2025

---

> ### Author Rebuttal · Authors · 2026-03-31
>
> Dear Reviewer M1Ek,
>
> We are grateful for the thoughtful comments. We hope our responses below address your concerns.
>
> **W1&Q1: Freezing $A$ in noisy branch**
>
> $A$ is structurally more robust than $B$, making it ideal as a shared subspace. Freezing $A$ in the noisy branch is a design choice; since $A$ already learns robust representations in the clean branch, further updates on noisy samples yielded negligible gains. This ensures $B^n$ absorbs noise without perturbing the shared subspace or adding computation.
>
> **W2: Novelty of the paper**
>
> Unlike prior works focusing on general knowledge or task interference, we specifically investigate the robustness asymmetry of LoRA under label noise. We justify this property empirically and theoretically. Figure 1 shows $A$ remains stable across clean and noisy labels while $B$ diverges. Appendix B provides a gradient-level explanation, noise perturbs $B$ via the raw error term, but affects $A$ only through the projected signal $B^\top \delta g$, structurally limiting corruption.
>
> FDLoRA exploits this by isolating noise into branch $B^n$ while preserving $A$ as a shared subspace. Our design uses discrepancy-based partitioning and negative learning to exploit noisy samples. By independently aggregating $B^c$ and $B^n$, FDLoRA enables effective sharing of informative low-rank components while preventing noise propagation. Our novelty lies in identifying and exploiting this structural asymmetry for robust federated fine-tuning.
>
> **W3: Problem and solution in the paper**
>
> FDLoRA is designed for the instability of federated LoRA under noise, rather than being a direct adaptation of centralized methods. LoRA's low-rank constraint makes local models prone to overfitting noise, which is then amplified via aggregation. FDLoRA decouples local adaptation into clean ($B^c$) and noisy ($B^n$) branches. $B^c$ learns robust semantics, while $B^n$ absorbs noise, preventing contamination of the shared representation. We then aggregate $B^c$ and $B^n$ independently to prevent noise propagation.
>
> **W4: Differences from FedSA**
>
> - FedSA aggregates $A$ while keeping $B$ local for clean heterogeneous FL.
> - FDLoRA aggregates decoupled $B^c$ and $B^n$ while keeping $A$ local for noisy heterogeneous FL.
>
> **W5: Discussion on generative LLM fine-tuning**
>
> While we focus on classification, LoRA's asymmetry theoretically applies to generative modeling. Our priority is providing a rigorous mechanistic explanation within a label-noise framework. We will add a discussion to include generative fine-tuning as a direction for future work.
>
> **W6&Q2: Sensitivity studies and heterogeneous noise analysis**
>
> We conduct experiments on heterogeneous noise scenarios. We assign varying noise rates of {0.1, 0.2, 0.3, 0.4, 0.5} across 10 clients.
>
> *Table R1: Performance comparison under heterogeneous label noise distributions.*
> |Method|FedLR|FFALoRA|RoLoRA|FLoRA|FlexLoRA|LoRA-FAIR|SLoRA|FedSA-LoRA|RFedLR|FDLoRA|
> |-|-|-|-|-|-|-|-|-|-|-|
> |Pairlflip|71.06|70.67|69.16|69.85|49.98|70.83|75.56|64.14|77.07|**88.33**|
> |Symflip|83.09|79.22|83.64|80.87|53.02|83.14|86.46|70.92|85.97|**90.95**|
>
> We evaluate FDLoRA in a larger-scale federated setting with 100 clients and a 10% participation rate per round.
>
> *Table R2: Performance comparison in a large-scale federated setting under 40% label noise.*
> |Method|FedLR|FFALoRA|RoLoRA|FLoRA|FlexLoRA|LoRA-FAIR|SLoRA|FedSA-LoRA|RFedLR|FDLoRA|
> |-|-|-|-|-|-|-|-|-|-|-|
> |Pairflip|45.39|42.61|43.09|45.28|14.62|43.95|36.92|42.62|46.62|**47.75**|
> |Symflip|64.45|63.68|65.18|66.22|19.88|65.02|54.67|63.66|67.36|**69.72**|
>
> We test the sensitivity of our method to the LoRA rank. The scaling factor $\alpha$ is adjusted proportionally with the rank to maintain update magnitude.
>
> *Table R3: Sensitivity analysis to different LoRA ranks under 40% Symflip label noise.*
> |$r$|2|4|8|16|32|
> |-|-|-|-|-|-|
> |Acc|90.80|90.92|90.98|91.10|90.85|
>
> We conduct experiments across three different random seeds, reporting Mean $\pm$ Standard deviation.
>
> *Table R4: Ablation study of FDLoRA components across 3 random seeds.*
> |DLF|NNL|$\mu=0.2$ (Pairflip)|$\mu=0.2$ (Symflip)|$\mu=0.4$ (Pairflip)|$\mu=0.4$ (Symflip)|
> |-|-|-|-|-|-|
> |||$88.22\pm0.14$|$88.53\pm0.09$|$84.99\pm0.76$|$88.56\pm0.28$|
> |$\checkmark$||$90.47\pm0.14$|$90.63\pm0.06$|$88.00\pm1.10$|$89.94\pm0.16$|
> |$\checkmark$|$\checkmark$|$91.43\pm0.09$|$91.15\pm0.19$|$88.93\pm1.03$|$90.98\pm0.25$|
>
> Due to the time constraints of the current rebuttal window, we will provide the results on local epochs in the next discussion phase.
>
> **Q3: Communication and memory overhead**
>
> We quantify the overhead of FDLoRA against FedLR, reporting average metrics across all stages.
>
> *Table R5: Quantification of the per-round communication cost (MB), computation cost (s), and the trainable parameters (%) relative to FFT.*
> ||Communication Cost (MB)|Computation Cost (s)|Trainable Parameter (%)|
> |-|-|-|-|
> |FedLR|0.8691|755.99|0.3503|
> |FDLoRA|0.8619|646.93|0.4351|

---

> > ### Author Rebuttal · Reviewer_M1Ek · 2026-04-02
> >
> > I thank the authors for the additional experiments. However, I still have concerns about the rank sensitivity analysis (Table R3). The results only show stability of the proposed method, without comparison to baselines across ranks. A convincing analysis should demonstrate consistent gains over baselines under different ranks. Moreover, the nearly flat trend is somewhat unexpected compared to prior FL LoRA works. I also encourage the authors to include a clearer comparison with [1],[2],[3] in the final version.
> >
> > I would be open to raising my score if these concerns are adequately addressed.

---

> > > ### Author Response · Authors · 2026-04-05
> > >
> > > Thank you for the constructive feedback. We agree that a convincing rank sensitivity analysis should include baseline comparisons across different ranks. Following your suggestion, we add results for all baselines under rank $r \in \lbrace 2,4,8,16,32 \rbrace$. The results in Table R6 show that FDLoRA consistently outperforms all baselines across different low-rank capacities. We believe this is because FDLoRA does not mainly rely on increasing adapter capacity, but on separating robust feature learning from noise modeling. Therefore, once the rank is sufficient to capture the task-relevant low-rank subspace, increasing it further brings only limited gains, leading to a relatively flat trend. We will clarify this point in the final version.
> > >
> > > *Table R6: Rank sensitivity analysis under different LoRA ranks $r$, with the scaling factor $\alpha$ adjusted proportionally to $r$.*
> > > |Rank|2|4|8|16|32|
> > > |-|-|-|-|-|-|
> > > |FedLR|86.39|85.26|86.47|85.43|85.85|
> > > |FFALoRA|79.61|81.34|82.45|82.49|83.28|
> > > |RoLoRA|85.19|85.48|85.71|85.86|86.12|
> > > |FLoRA|77.45|76.99|78.07|79.25|80.24|
> > > |FlexLoRA|85.46|85.80|85.05|86.68|86.19|
> > > |LoRA-FAIR|85.75|85.56|85.00|84.67|84.90|
> > > |SLoRA|86.31|87.18|87.63|88.16|88.46|
> > > |FedSA-LoRA|78.82|78.00|78.19|75.74|75.73|
> > > |RFedLR|82.48|83.73|82.86|84.38|83.51|
> > > |FDLoRA|**90.80**|**90.92**|**90.98**|**91.10**|**90.85**|
> > >
> > > We will incorporate a clearer discussion of prior work into the final version. These works mainly study the functional roles of LoRA factors in terms of general or shared knowledge,and task- or domain-specific knowledge.  Our work focuses on the robustness asymmetry of LoRA factors under federated noisy setting. We show both empirically and theoretically that $A$ remains more stable under noisy supervision, whereas $B$ is more directly affected by corrupted labels. Based on this observation, FDLoRA is designed to isolate noise into the noisy branch while preserving robust feature learning in the clean branch, which is different from prior methods.
> > >
> > > As promised, we now complete the ablation study on the number of local epochs, with $E \in \lbrace 1,2,3,4\rbrace$. As shown in Table R7, FDLoRA achieves the best performance at $E=1$, and the accuracy decreases as $E$ increases. We attribute this to the fact that larger local epochs lead to stronger local overfitting to noisy labels, while more frequent global aggregation provides a regularizing effect. Therefore, $E=1$ is the optimal choice in our setting.
> > >
> > > *Table R7: Ablation study on the number of local epochs $E$. We report the average test accuracy (\%) of FDLoRA.*
> > > |$E$|1|2|3|4|
> > > |-|-|-|-|-|
> > > |Acc|90.91|90.42|87.86|84.16|
> > >
> > > We hope these additional results and clarifications address your concerns.

---

### Official Review · Reviewer_JfqA · 2026-03-11

**Soundness:** 3
**Presentation:** 2
**Significance:** 3
**Originality:** 3
**Overall Recommendation:** 4
**Confidence:** 4

**Summary:**

This paper studies robust federated fine-tuning of large pre-trained models under label noise. It analyzes the different sensitivities of LoRA’s low-rank components to noisy labels and proposes Federated Decoupled LoRA (FDLoRA), which separates robust feature learning from noise modeling through a dual-branch design with a noise-aware training strategy. Experiments show that the proposed method achieves improved performance in noisy federated settings.

**Compliance With Llm Reviewing Policy:**

Affirmed.

**Final Justification:**

My concerns have been adequately addressed.

**Key Questions For Authors:**

1. In Figure 1, the paper reports similarity comparisons between the two LoRA matrices across different tasks. Could the authors clarify how this similarity is computed?
2. The method relies on the observation that the down-projection matrix learns robust representations while the up-projection matrix tends to fit noisy labels. Could the authors provide additional empirical or theoretical evidence supporting this claim across different architectures or datasets?
3. The experiments mainly evaluate synthetic label noise settings. How sensitive is the proposed method to different types of noise (e.g., instance-dependent noise) and to varying levels of client data heterogeneity in federated settings?
4. Does the proposed dual-branch training and asymmetric aggregation introduce additional computational or communication overhead compared to standard federated LoRA training? If so, it would be helpful to quantify this overhead.

**Limitations:**

Please address the questions in Weaknesses and Key Questions.

**Strengths And Weaknesses:**

*Strengths:*
- The paper studies robust federated fine-tuning of large pre-trained models under label noise, a setting that is relevant for deploying parameter-efficient adaptation methods in distributed environments.
- The method is motivated by the asymmetric behavior of the two LoRA matrices and proposes a decoupled design to separate feature learning from noise modeling, providing an interesting perspective for improving robustness.
- Experimental results show performance improvements over several baselines in noisy federated settings.

*Weaknesses:*
- The justification for the robustness asymmetry between the two LoRA matrices remains largely intuitive, and stronger theoretical analysis or broader empirical validation would strengthen the claim.
- The experimental evaluation is conducted on a limited set of datasets and client configurations, which may not fully reflect diverse federated environments.
- Some parts of the method description are relatively dense, making the training procedure and module interactions difficult to follow.
- The paper provides limited analysis explaining why the proposed design improves robustness under label noise. Additional analysis or visualization could help clarify the underlying mechanism.

---

> ### Author Rebuttal · Authors · 2026-03-31
>
> Dear Reviewer JfqA,
>
> Thanks for your careful review and valuable suggestions. We hope our response addresses your concerns.
>
> **W1&Q2: Verifying robustness asymmetry of LoRA**
>
> To verify asymmetry as a structural property, we evaluate CV (ViT/CIFAR-100) and NLP (RoBERTa/20ng) tasks. Table R1 shows that across architectures, layer-wise average cosine similarity between clean and noisy models remains stable for $A$, while $B$ diverges, confirming $B$ fits noisy labels.
>
> *Table R1: Layer-wise cosine similarity between clean and noisy LoRA matrices.*
> ||ViT (CIFAR-100)|RoBERTa (20 Newsgroups)|
> |-|-|-|
> |Query A|0.9996|0.9729|
> |Query B|0.7458|0.4742|
> |Value A|0.9916|0.9837|
> |Value B|0.8035|0.6258|
>
> **W2: Evaluation on diverse datasets and client configurations**
>
> To validate generalization, we tested Tiny-ImageNet at 0.4 noise rate. Table R2 shows FDLoRA significantly outperforms all baselines.
>
> *Table R2: Performance comparison on the Tiny-ImageNet dataset  under 40% label noise.*
> |Method|FedLR|FFALoRA|RoLoRA|FLoRA|FlexLoRA|LoRA-FAIR|SLoRA|FedSA-LoRA|RFedLR|FDLoRA|
> |-|-|-|-|-|-|-|-|-|-|-|
> |Pairflip|61.81|59.77|61.78|62.68|50.17|63.44|65.29|57.99|62.68|**87.07**|
> |Symflip|87.33|83.81|87.25|86.84|71.73|87.50|89.08|78.89|88.09|**90.64**|
>
> We evaluate FDLoRA with 100 clients and 10% per-round participation. Clients unselected  in the first $T_0+T_1$ rounds  perform a single probability estimation step to establish data partitions before NNL.
>
> *Table R3: Performance comparison in a large-scale federated setting under 40% label noise.*
> |Method|FedLR|FFALoRA|RoLoRA|FLoRA|FlexLoRA|LoRA-FAIR|SLoRA|FedSA-LoRA|RFedLR|FDLoRA|
> |-|-|-|-|-|-|-|-|-|-|-|
> |Pairflip|45.39|42.61|43.09|45.28|14.62|43.95|36.92|42.62|46.62|**47.75**|
> |Symflip|64.45|63.68|65.18|66.22|19.88|65.02|54.67|63.66|67.36|**69.72**|
>
> **W3: Clarification on training procedure**
>
> FDLoRA evolves across three stages during $T$ communication rounds:
>
> Local Fine-Tuning:
> - Dual-Branch Warm-Up. Partition data by loss. Low-loss samples update $A$ and $B^c$ and high-loss update $B^n$.
> - Decoupled LoRA Fine-Tuning. Route data via loss discrepancy and EMA. $B^c$ optimizes decision boundaries while $B^n$ fits local noise.
> - Noisy Branch Negative Learning. For noisy data, $B^c$ provides soft targets and uses KL divergence to force branch divergence.
>
> Asymmetric Global Aggregation: Matrix $A$ is kept local. $B^c$ and $B^n$ are aggregated independently. $B^n$ is discarded during inference, which relies solely on $B^c$.
>
> **W4: Explanation of robustness mechanism**
>
> As shown in Figure 1 and Appendix B, $A$ remains stable while $B$ is directly corrupted by noise gradients. FDLoRA exploits this by decoupling $B$. The clean branch $B^c$ learns robust semantics from low-loss samples, while the noisy branch $B^n$ captures noise patterns from high-loss samples. This prevents parameter conflicts in a single adapter. Furthermore, the negative learning term explicitly stops $B^c$ from drifting toward noisy predictions. We will clarify this mechanism in the revision and add a visualization of branch loss distributions to illustrate how the two branches specialize under label noise.
>
> **Q1: Computation of LoRA matrix similarity**
>
> We fine-tune two identical models on clean and noisy data, extracting $A$ and $B$ from Query and Value modules across layers. After flattening them into vectors, we compute cosine similarity between corresponding vectors to quantify parameter deviation caused by label noise.
>
> **Q3: Robustness to instance-dependent noise and severe data heterogeneity**
>
> To evaluate FDLoRA under more realistic noise distributions, we conduct experiments with instance-dependent noise.
>
> *Table R4: Performance comparison under instance-dependent noise with a noise rate of 0.4.*
> |Method|FedLR|FFALoRA|RoLoRA|FLoRA|FlexLoRA|LoRA-FAIR|SLoRA|FedSA-LoRA|RFedLR|FDLoRA|
> |-|-|-|-|-|-|-|-|-|-|-|
> |Acc|60.19|60.53|62.06|59.40|42.01|59.81|72.44|47.25|71.11|**88.23**|
>
> To demonstrate the robustness of FDLoRA under severe data heterogeneity, we evaluate an extreme Non-IID setting (Dirichlet $\beta=0.1$).
>
> *Table R5: Performance comparison under severe data heterogeneous scenario when the noise rate is 0.4.*
> |Method|FedLR|FFALoRA|RoLoRA|FLoRA|FlexLoRA|LoRA-FAIR|SLoRA|FedSA-LoRA|RFedLR|FDLoRA|
> |-|-|-|-|-|-|-|-|-|-|-|
> |Pairlflip|56.24|52.77|56.08|54.68|33.79|55.77|58.76|47.25|57.15|**86.78**|
> |Symflip|74.86|61.28|74.40|63.50|33.54|75.36|80.07|53.46|72.99|**89.81**|
>
> **Q4: Computational and communication overhead**
>
> To quantify the efficiency of FDLoRA, we compare its per-round communication cost, computation cost, and trainable parameters against FedLR, reporting the average metrics across all stages.
>
> *Table R6: Quantification of the per-round communication cost (MB), per-round computation cost (s), and the trainable parameters (%) relative to FFT.*
> ||Communication Cost (MB)|Computation Cost (s)|Trainable Parameter (%)|
> |-|-|-|-|
> |FedLR|0.8691|755.99|0.3503|
> |FDLoRA|0.8619|646.93|0.4351|

---

> > ### Author Rebuttal · Reviewer_JfqA · 2026-04-04
> >
> > Thank you to the authors for the detailed explanations and the additional experimental validation provided in response to my questions. These supplementary results and clarifications significantly enhance the overall persuasiveness of the paper.
> > I note your commitment to including the branch loss distribution visualizations in the subsequent version. The core research conclusions have been strengthened by your rebuttal.
> > In light of these improvements and the effort taken to address my concerns, I will be raising my score.

---

### Official Review · Reviewer_ULKC · 2026-03-13

**Soundness:** 2
**Presentation:** 2
**Significance:** 2
**Originality:** 2
**Overall Recommendation:** 4
**Confidence:** 4

**Summary:**

The paper studies the issue of the federated fine-tuning of LLM using LoRA under label noise. the main claim of this paper is that the LoRA A matrix is stable while B is more noise unstable under lable noise. The paper introduces FDLoRA a dual branch method where B is aggregated and A stays local.

**Compliance With Llm Reviewing Policy:**

Affirmed.

**Final Justification:**

The rebuttal addressed some of main concerns therefore i will increase my score from weak reject to weak accept.

**Key Questions For Authors:**

Check Weaknesses

**Limitations:**

Yes

**Strengths And Weaknesses:**

**Strenghts**

* The paper tackle an important problem of federated fine-tuning of LLM using Lora.

* The paper shows a broad comparison against many FL Lora baslines.

* The routing mechansim which is basccialy loss difference + EMA is straightforward and easy to implment.

* The paper is well-written and easy to follow


**Weaknesses**

* The theory is limited to gradient level observation

* I belive that aggregating noisy branch B across clients may propagate errors globally

* it's unclear whether FDLoRA is stable under partial participation

* In Table 1, FDLoRA reported 88.30% on cifar100 with 40% pairflip for 10 client under non-iid setting. This is a unusall gain and there is no error bars or seeds in order to interpert the results which rasie concerns about robustness.

* only CIRA100 dataset and small scale NLP setup are tested

* It's unclear to me which B factors are aggregated and why aggregating $B^n$ is beneficial depite being noisy ?

* I believe that the KL term on the noisy samples is unbounded

---

> ### Author Rebuttal · Authors · 2026-03-31
>
> Dear Reviewer ULKC,
>
> We appreciate your thoughtful review and insightful comments. We hope the following responses help clarify the issues you raised.
>
>  **W1: Theory and empirical corroboration**
>
> While our proof focuses on gradients, they are the fundamental pathway for noise corruption. As Appendix B details, noise directly impacts the up-projection matrix $B$, but its influence on the down-projection matrix $A$ is structurally restricted by $B^\top$ to a low-rank subspace. Crucially, this theory is corroborated empirically. Figure 1 shows that $A$ matrices trained on clean and noisy data maintain highly cosine similarity, whereas $B$ matrices severely diverge to fit noisy targets. This synergy of gradient-level theory and empirical validation fundamentally motivates decoupled architecture of FDLoRA.
>
> **W2: Clarification on global error propagation of noisy branch $B^n$**
>
> Although $B^n$ aggregates noisy patterns, it does not contaminate the final model as this information remains strictly isolated. The noisy branch only facilitates sample separation and negative learning during training. During inference, $B^n$ is discarded and the model relies exclusively on the clean branch $B^c$, ensuring global noise does not impact performance.
>
> **W3: Effectiveness under partially participating federated settings**
>
> We evaluate FDLoRA with 100 total clients and 10% per-round participation. Clients unselected during the initial $T_0+T_1$ rounds perform a single probability estimation step to establish data partitions before NNL. As shown in Table R1, FDLoRA achieves 47.75% accuracy under Pairflip and 69.72% under Symflip noise, outperforming RFedLR. These results prove FDLoRA remains stable and superior under partial participation.
>
> *Table R1: Performance comparison in a partially participating federated setting under 40% label noise.*
> |Method|FedLR|FFALoRA|RoLoRA|FLoRA|FlexLoRA|LoRA-FAIR|SLoRA|FedSA-LoRA|RFedLR|FDLoRA|
> |-|-|-|-|-|-|-|-|-|-|-|
> |Pairflip|45.39|42.61|43.09|45.28|14.62|43.95|36.92|42.62|46.62|**47.75**|
> |Symflip|64.45|63.68|65.18|66.22|19.88|65.02|54.67|63.66|67.36|**69.72**|
>
> **W4: Variance Analysis of FDLoRA Components**
>
> To address stability concerns, we conducted experiments across three random seeds (0, 1, 42). The results (Table R2) show FDLoRA achieves consistently high performance. The significant gain under 40% Pairflip noise stems from our architecture. While structured noise corrupts the sensitive matrix $B$ and causes model collapse, our DLF isolates this noise into $B^n$. NNL then uses this isolated noise to push clean decision boundaries away from errors.
>
> *Table R2: Ablation study of FDLoRA components across 3 random seeds.*
> |DLF|NNL|$\mu=0.2$ (Pairflip)|$\mu=0.2$ (Symflip)|$\mu=0.4$ (Pairflip)|$\mu=0.4$ (Symflip)|
> |-|-|-|-|-|-|
> |||$88.22\pm0.14$|$88.53\pm0.09$|$84.99\pm0.76$|$88.56\pm0.28$|
> |$\checkmark$||$90.47\pm0.14$|$90.63\pm0.06$|$88.00\pm1.10$|$89.94\pm0.16$|
> |$\checkmark$|$\checkmark$|$91.43\pm0.09$|$91.15\pm0.19$|$88.93\pm1.03$|$90.98\pm0.25$|
>
> **W5: Supplementary evaluation on Tiny-ImageNet**
>
> To validate generalization, we test Tiny-ImageNet at 0.4 noise rate. As shown in Table R3, FDLoRA significantly outperforms all baselines. Notably, under Pairflip noise, FDLoRA reached 87.07% accuracy, far exceeding the 62.68% of FLoRA and RFedLR. These results prove FDLoRA is effective and scales successfully to larger datasets.
>
> *Table R3: Performance comparison on the Tiny-ImageNet dataset  under 40% label noise setting.*
> |Method|FedLR|FFALoRA|RoLoRA|FLoRA|FlexLoRA|LoRA-FAIR|SLoRA|FedSA-LoRA|RFedLR|FDLoRA|
> |-|-|-|-|-|-|-|-|-|-|-|
> |Pairflip|61.81|59.77|61.78|62.68|50.17|63.44|65.29|57.99|62.68|**87.07**|
> |Symflip|87.33|83.81|87.25|86.84|71.73|87.50|89.08|78.89|88.09|**90.64**|
>
> **W6: Clarification on $B$ matrix aggregation**
>
> In FDLoRA, the server independently aggregates clean branches $B^c$ and noisy branches $B^n$. Aggregating $B^n$ constructs a global noise extractor that provides sharper loss discrepancy signals, enhancing sample separation accuracy during local training. Crucially, $B^n$ serves only as an auxiliary component for partitioning and negative learning. It is completely discarded during inference, ensuring the final deployed model remains uncontaminated.
>
> **W7: Clarification on the implemented negative KL loss**
>
> Thank you for pointing this out. We clarify that the implementation uses a stabilized version of the objective instead of the raw $-\mathrm{KL}(q_i^c \parallel q_i^n)$. In practice, the noisy branch acts as a detached reference where we compute the KL term using an $\epsilon$-floored and renormalized distribution $\tilde{q}_i^n = \mathrm{Normalize}(\max(q_i^n, \epsilon))$. Since every entry of $\tilde{q}_i^n$ is bounded away from zero, $\mathrm{KL}(q_i^c \parallel \tilde{q}_i^n)$ is upper-bounded by a finite constant and cannot diverge. We will revise the manuscript to explicitly reflect this implemented formulation.

---

> > ### Author Rebuttal · Reviewer_ULKC · 2026-04-02
> >
> > Thank you for the detailed and constructive rebuttal. I appreciate the authors’ effort in providing additional evidence, including the partial-participation results, multi-seed variance analysis, the Tiny-ImageNet evaluation, and the clarifications regarding the role of the noisy branch $B^n$ and the stabilized KL implementation. These additions address several of my main concerns and improve my confidence in the empirical robustness and practical clarity of the method.
> >
> > While I still think some of the theoretical aspects remain somewhat limited, the rebuttal substantially strengthens the paper overall. In particular, the additional experiments and implementation clarifications make the contribution more convincing to me. I will therefore increase my score.

---

> > > ### Author Response · Authors · 2026-04-02
> > >
> > > Thank you for the thoughtful and encouraging follow-up. We sincerely appreciate your positive reassessment and the increased score. We will incorporate these additions and clarifications into the final version to further improve the completeness and clarity of the paper.
> > >
> > > Our current theory is a mechanism-level analysis aimed at explaining why the two LoRA factors behave differently under noisy supervision. Appendix B shows that noise perturbs $B$ directly, while its effect on $A$ must pass through $B^\top$, and is thus restricted to a low-rank subspace. This structural explanation is corroborated by Figure 1, where $A$ learned from clean and noisy data remains highly aligned, whereas $B$ diverges substantially. This consistency between theory and empirics directly motivates FDLoRA's decoupled design and the asymmetric treatment of $A$ and $B$.
> > >
> > > We agree that the current theoretical analysis is focused in scope. We intentionally focus on the structural noise-propagation asymmetry between $A$ and $B$, since this is the most direct theoretical basis for our method. A broader treatment of the federated setting would require substantially stronger assumptions on training dynamics, client heterogeneity, and label noise. Therefore, we aim to provide a precise justification that is closely aligned with both Figure 1 and the key design choices of FDLoRA. We will clarify this scope more explicitly in the revision, and we also view strengthening the theory in this direction as an important avenue for our future work.

---

### Official Review · Reviewer_FXiq · 2026-03-13

**Soundness:** 3
**Presentation:** 3
**Significance:** 3
**Originality:** 3
**Overall Recommendation:** 4
**Confidence:** 4

**Summary:**

This paper proposes Federated Decoupled LoRA (FDLoRA) for robust federated fine-tuning under label noise. The key insight is a "robustness asymmetry" in LoRA's factorization: the down-projection matrix A learns stable features invariant to noise, while the up-projection matrix B is susceptible to fitting noise patterns. Based on this, FDLoRA employs a dual-branch architecture with a clean branch for robust feature learning and a noisy branch for noise modeling via negative learning. Only B matrices are aggregated globally while A remains local. Experiments on CIFAR-100 and 20 Newsgroups show FDLoRA significantly outperforms SOTA federated LoRA methods under various label noise settings.

**Compliance With Llm Reviewing Policy:**

Affirmed.

**Final Justification:**

I am maintaining my score of 4 (Weak Accept). While the authors' rebuttal provided a detailed response to my concerns regarding the scalability bottleneck, it did not fully convince me. The method still appears to exhibit a noticeable performance drop and lose some of its competitive edge in a realistic 100-client scenario. Additionally, as other reviewers have also noted, the observation regarding the asymmetry between LoRA's A and B matrices shares similarities with existing work. Given the remaining concerns, I am inclined to keep my current score.

**Key Questions For Authors:**

1. **Hyperparameter sensitivity**: The method relies on several hyperparameters including the loss thresholds ($\delta\_{low}$, $\delta\_{high}$), temperature $\tau$, and KL divergence weight $\eta$. How sensitive is the performance to these choices? Have you conducted ablation studies on different values, particularly for the threshold selection strategy? This would help assess the method's practicality in real-world deployment where clean validation data may not be available for hyperparameter tuning.

2. **Scalability and Non-IID realism**: The experiments use only K=10 clients for CIFAR-100. This small number of clients cannot adequately reflect the data heterogeneity (Non-IID) challenges in realistic federated learning scenarios, which typically involve hundreds or thousands of clients with much smaller local datasets. With fewer clients, the degree of Non-IID that can be simulated is limited, potentially masking the method's true robustness to severe data heterogeneity. How does the method perform with K=100+ clients? Would the clean/noisy sample identification remain effective with smaller local datasets? Empirical validation at realistic scale would strengthen the contribution.

3. **Why does asymmetric aggregation work for heterogeneous clients?** Section 3.5 states that "the high variance of the B matrix reflects diverse local semantic distributions" and aggregating B facilitates knowledge exchange. However, if clients have different label noise patterns, wouldn't aggregating B also aggregate the noise patterns? Please clarify why this doesn't negatively impact the global model.

4. **Relationship to FFA-LoRA and RFedLR**: FFA-LoRA (ICLR 2024) also freezes matrix A and updates only matrix B, motivated by aggregation error and communication efficiency. RFedLR (NeurIPS 2025) proposes sensitivity-aware robust tuning that identifies noise-sensitive parameters. How does FDLoRA's "robustness asymmetry" observation differ from these prior approaches? Specifically, (a) Is the observation that A is more robust to label noise than B a novel contribution of this work, or has it been previously identified? (b) How do the motivations and technical approaches differ from FFA-LoRA and RFedLR? Clarifying these distinctions would help better position the novelty of this work.

**Limitations:**

yes

**Strengths And Weaknesses:**

**Strengths**

1. **Theoretical and empirical insight into LoRA structure**: The paper provides a solid theoretical foundation (Lemma B.1) and empirical evidence (Figure 1) for the "robustness asymmetry" phenomenon—where the down-projection matrix A extracts stable general features invariant to label noise while the up-projection matrix B is susceptible to fitting noise patterns. This observation provides a fresh perspective on designing robust federated learning methods.

2. **Well-designed method**: The proposed three-phase training procedure (warm-up, decoupled fine-tuning, negative learning) is carefully designed and logically motivated. The dual-branch architecture that leverages LoRA's structural asymmetry is a creative solution that effectively decouples robust feature learning from noise modeling.

3. **Comprehensive experimental evaluation**: The experiments cover both vision (CIFAR-100) and NLP (20 Newsgroups) tasks with multiple noise types (Pairflip, Symmetric) and ratios ($\mu$= 0.2, 0.4). The strong empirical results (e.g., 88.30% vs ~56-63% for baselines under $\mu$=0.4 Pairflip noise) demonstrate substantial practical impact.

4. **Clear presentation**: The paper is generally well-written with logical flow from observation to method design. Figure 2 effectively illustrates the framework architecture, and the theoretical analysis in Appendix B provides deeper insight into why the asymmetry occurs.

**Weaknesses**

1. **Limited analysis of hyperparameter sensitivity**: The sample selection strategy relies heavily on loss-based thresholds ($\delta\_{low}$, $\delta\_{high}$), but the sensitivity of these hyperparameters is not thoroughly analyzed. Additionally, the paper lacks discussion on why certain hyperparameter choices ($\lambda$=0.8, $\eta$=0.01 for Pairflip, $\eta$=0.3 for symmetric) work better than others.

2. **Scale and scope limitations**: The evaluation uses only K=10 clients for CIFAR-100 and K=3 clients for 20 Newsgroups (not "selecting 10 clients per round" but "only 10 clients total"). This small-scale setting cannot adequately reflect the data heterogeneity (Non-IID) challenges in realistic federated learning scenarios, which typically involve hundreds or thousands of clients with much smaller local datasets. The method is also designed specifically for fine-tuning pre-trained models; its applicability to training from scratch is not discussed.

3. **Need for better differentiation from related work**: While the robustness asymmetry observation appears to be novel, the paper could better clarify its distinction from closely related works. FFA-LoRA (ICLR 2024) also treats A and B differently (freezing A, updating B), though motivated by aggregation error rather than noise robustness. RFedLR (NeurIPS 2025) addresses parameter sensitivity to noise but does not specifically identify the A/B asymmetry in the context of label noise. The paper should more explicitly articulate these differences to establish its novelty.

---

> ### Author Rebuttal · Authors · 2026-03-31
>
> Dear Reviewer FXiq,
>
> Thanks for thevaluable feedback and constructive comments. We hope our detailed responses below address your concerns.
>
> **W1&Q1: Analysis of hyperparameter sensitivity**
>
> We conduct comprehensive ablation studies on the smoothing factor $\lambda$, temperature $\tau$, and negative learning weight $\eta$.
>
> For the smoothing factor $\lambda$ in EMA, the performance remains remarkably stable between 0.5 and 0.9. Table R1 demonstrates that the framework is robust to $\lambda$.
>
> *Table R1: Parameter analysis for the smoothing factor $\lambda$ in EMA under 40% Symflip noise.*
> |$\lambda$|0.5|0.6|0.7|0.8|0.9|
> |-|-|-|-|-|-|
> |Acc|90.87|90.86|90.90|90.92|90.91|
>
> Evaluating the temperature parameter $\tau$ from 0.5 to 0.9 yields consistent accuracy around 90.8% (Table R2). The method maintains optimal performance across different temperature scales.
>
> *Table R2: Parameter analysis for the temperature parameter $\tau$ under 40% Symflip noise.*
> |$\tau$|0.5|0.6|0.7|0.8|0.9|
> |-|-|-|-|-|-|
> |Acc|90.81|90.82|90.91|90.89|90.92|
>
> Regarding the negative learning weight $\eta$, accuracy remains stable across $\eta \in [0.1, 0.5]$ under Symflip noise. Conversely, Pairflip noise requires a smaller optimal weight of $\eta=0.01$, thereby preventing over-suppressing the clean branch against structured, concentrated perturbations.
>
> *Table R3: Parameter analysis for the negative learning weight $\eta$ under 40% Symflip noise.*
> |$\eta$|0.1|0.2|0.3|0.4|0.5|
> |-|-|-|-|-|-|
> |Acc|90.86|90.88|90.92|90.92|90.90|
>
> *Table R4: Parameter analysis for the negative learning weight $\eta$ under 40% Pairflip noise.*
> |$\eta$|0.01|0.04|0.07|0.1|
> |-|-|-|-|-|
> |Acc|88.30|87.14|86.70|84.43|
>
> $\delta_{low}$ is a dynamic quantile threshold. During warm-up, $\delta_{low}$ decreases linearly from the 100th to the 10th percentile of clean branch losses to select the cleanest samples. Meanwhile, $\delta_{high}$ remains fixed at the 90th percentile to isolate the noisy subset.
>
> **W2&Q2: Effectiveness under large-scale FL settings**
>
> We evaluate FDLoRA with 100 clients and 10% per-round participation. Clients unselected  in the first $T_0+T_1$ rounds  perform a single probability estimation step to establish data partitions before applying NNL. Table R5 shows FDLoRA maintains its advantage in large-scale federated settings.
>
> *Table R5: Performance comparison in a large-scale federated setting under 40% label noise.*
> |Method|FedLR|FFALoRA|RoLoRA|FLoRA|FlexLoRA|LoRA-FAIR|SLoRA|FedSA-LoRA|RFedLR|FDLoRA|
> |-|-|-|-|-|-|-|-|-|-|-|
> |Pairflip|45.39|42.61|43.09|45.28|14.62|43.95|36.92|42.62|46.62|**47.75**|
> |Symflip|64.45|63.68|65.18|66.22|19.88|65.02|54.67|63.66|67.36|**69.72**|
>
> To demonstrate the robustness of FDLoRA under severe data heterogeneity, we evaluate an extreme Non-IID setting, with the Dirichlet concentration parameter set to $\beta=0.1$. Table R6 shows FDLoRA consistently outperforms all baselines.
>
> *Table R6: Performance comparison under severe data heterogeneous scenario when the noise rate is 0.4.*
> |Method|FedLR|FFALoRA|RoLoRA|FLoRA|FlexLoRA|LoRA-FAIR|SLoRA|FedSA-LoRA|RFedLR|FDLoRA|
> |-|-|-|-|-|-|-|-|-|-|-|
> |Pairlflip|56.24|52.77|56.08|54.68|33.79|55.77|58.76|47.25|57.15|**86.78**|
> |Symflip|74.86|61.28|74.40|63.50|33.54|75.36|80.07|53.46|72.99|**89.81**|
>
> While this work focuses on LoRA-based federated fine-tuning of pre-trained models, future research will extend it to training from scratch.
>
> **W3&Q4: Clarification on novelty and related work**
>
> (a) To the best of our knowledge, we are the first to investigate the robustness asymmetry of LoRA under label noise. Empirically, $A$ stably captures input features regardless of label correctness, while $B$ significantly distorts to fit noisy targets. Theoretically, noise directly impacts $B$’s gradients, while its influence on $A$ is restricted by $B^\top$ to a bounded low-rank subspace, motivating FDLoRA.
>
> (b) While FFA-LoRA freezes $A$ and updates $B$ to mitigate aggregation bias, our study reveals $B$ overfits noisy targets. FDLoRA isolates noise into a separate noisy branch. Unlike RFedLR, which treats LoRA adapters as generic weights and requires a clean proxy dataset, FDLoRA requires no proxy data, instead exploits the robustness asymmetry between $A$ and $B$ to separate noisy samples via loss differences.
>
> **Q3: Clarification on noise propagation in asymmetric aggregation**
>
> Standard federated LoRA aggregates a single matrix $B$, propagating  noise. FDLoRA resolves this using a dual-branch architecture that independently aggregates the clean branch $B^c$ and noisy branch $B^n$, enabling knowledge sharing without noise propagation.
>
> Aggregating $B^c$ builds a noise-free global consensus, mitigating Non-IID effects. Aggregating $B^n$ creates a global noise extractor, improving local sample separation. Crucially, $B^n$ is an auxiliary training component discarded during inference. Since the model then relies solely on $B^c$, global noise patterns cannot degrade final performance.

---

> > ### Author Rebuttal · Reviewer_FXiq · 2026-04-03
> >
> > I appreciate the authors' efforts during the rebuttal phase. However, I have decided to maintain my score. First, the newly provided experiments with 100 clients (Table R5) reveal a severe performance degradation, wherein the proposed method largely loses its empirical advantage over the baselines. This raises significant concerns regarding its scalability and practical applicability in realistic federated settings. Second, as highlighted during the cross-reviewer discussions, the core observation regarding the asymmetry of the A and B matrices is not fundamentally novel, having been documented in prior literature. Consequently, I will maintain my current score of 4.

---

> > > ### Author Response · Authors · 2026-04-03
> > >
> > > Thank you for the thoughtful and valuable follow-up. Regarding Table R5, we would like to clarify that the 100 client setting is more challenging. Each client has much less local data, the non-IID level is higher, and only 10% of clients participate in each round. Under such a setting, all methods suffer clear performance degradation. In this context, FDLoRA still achieves the best performance under both Pairflip and Symflip noise.
> > >
> > > Prior works mainly discuss $A$ and $B$ asymmetry in terms of shared and personalized knowledge, task interference, or heterogeneous adaptation.  However, our contribution is to analyze this asymmetry from the perspective of robustness against label noise in federated fine-tuning. Figure 1 shows that $A$ remains highly similar when trained with clean and noisy labels, while $B$ changes substantially, indicating much stronger noise sensitivity in $B$. Appendix B further explains this asymmetry at the gradient level: label noise perturbs $B$ through the raw error term, but affects $A$ only through the projected signal $B^\top \delta g$, which structurally limits the influence of corrupted supervision on $A$. Based on this noise-specific robustness characterization, we design FDLoRA to preserve the stable feature subspace in $A$, isolate noise modeling into the noisy branch $B^n$, perform discrepancy-based noisy partition, and exploit noisy samples through negative learning. By independently aggregating the clean branch $B^c$ and noisy branch $B^n$, FDLoRA enables informative sharing of $B$ across clients while avoiding contamination of the inference branch. Therefore, the contribution of our work lies not in a generic claim that $A$ and $B$ are different, but in showing how their robustness asymmetry under label noise can be analyzed and exploited for robust federated LoRA fine-tuning.

---

### Decision · Program_Chairs · 2026-04-30

**Decision:**

Accept (regular)

**Comment:**

**Summary as I observed.**
In this paper, the authors propose FDLoRA, a federated fine-tuning framework aimed at improving the robustness of parameter-efficient adaptation in the presence of label noise. The core idea is built on an observed asymmetry between the two LoRA projection matrices, which motivates a dual-branch design that focuses on learning robust features and capturing noise-related patterns. It also combines this design with asymmetric aggregation in federated optimization, showing consistent gains across both vision and NLP tasks under a range of noisy federated settings. The paper receives final scores of 4/4/4/4.

**Strengths as I observed.**
This paper focuses on robust federated fine-tuning of large pre-trained models under label noise, which is a practically relevant setting, and the paper tackles it in a way that is both well motivated and technically coherent. In particular, the observation that the two LoRA components behave differently under noise provides a reasonable foundation for the proposed design. The experimental evaluation covers multiple noise types, heterogeneous settings, and both vision and NLP benchmarks, showing clear improvements over several strong baselines. I also found the rebuttal helpful. The additional results on scalability, heterogeneous noise, and rank sensitivity address several of the concerns raised in the initial reviews and increase my confidence that the method is not overly fragile to changes in setup.

**Weaknesses as I observed.**
First, the novelty of the asymmetry observation could be presented more clearly with respect to prior LoRA-based and federated fine-tuning methods. Second, although the experiments are fairly extensive, some reviewers were right to question how far the current evaluation goes toward reflecting truly large-scale or highly realistic federated deployments. Third, the theoretical support for the robustness claims is still somewhat limited relative to the breadth of the empirical conclusions. In the rebuttal, the authors responded constructively by adding additional experiments, sensitivity analyses, and clarifications on the method's relation to existing work. For the final version, I would encourage the authors to make the distinction from the closest related methods even more explicit and incorporate the scalability and sensitivity results into the main paper.

Following the rebuttal, all reviewers recommended acceptance. After reviewing the paper, the reviewers' comments, and the authors' responses, I am convinced that the strengths of the work outweigh its weaknesses. I therefore support acceptance.